# A cyclic peptide toolkit reveals mechanistic principles of peptidylarginine deiminase IV regulation

M. Teresa Bertran [1,11], Robert Walmsley [2,11], Thomas Cummings [2,3], Iker Valle Aramburu[4], Donald J. Benton [5], Rocio Mora Molina [2], Jayalini Assalaarachchi [2], Maria Chasampalioti [2], Tessa Swanton [4], Dhira Joshi [6], Stefania Federico[6], Hanneke Okkenhaug[7], Lu Yu [8], David Oxley[8], Simon Walker[7], Venizelos Papayannopoulos [4], Hiroaki Suga [9], Maria A. Christophorou [2,3] ✉ & Louise J. Walport [1,9,10] ✉

Peptidylarginine deiminase IV (PADI4, PAD4) deregulation promotes the development of autoimmunity, cancer, atherosclerosis and age-related tissue fibrosis. PADI4 additionally mediates immune responses and cellular reprogramming, although the full extent of its physiological roles is unexplored. Despite detailed molecular knowledge of PADI4 activation in vitro, we lack understanding of its regulation within cells, largely due to a lack of appropriate systems and tools. Here, we develop and apply a set of potent and selective PADI4 modulators. Using the mRNA-display-based RaPID system, we screen >10^{12} cyclic peptides for high-affinity, conformation-selective binders. We report PADI4_3, a cell-active inhibitor specific for the active conformation of PADI4; PADI4_7, an inert binder, which we functionalise for the isolation and study of cellular PADI4; and PADI4_11, a cell-active PADI4 activator. Structural studies with PADI4_11 reveal an allosteric binding mode that may reflect the mechanism that promotes cellular PADI4 activation. This work contributes to our understanding of PADI4 regulation and provides a toolkit for the study and modulation of PADI4 across (patho)physiological contexts.

The peptidylarginine deiminase enzymes (PADIs, or PADs, herein referred to as PADIs) catalyse the post-translational conversion of protein arginine residues to non-coded citrulline, in a process termed citrullination or peptidylarginine deimination[1]. In mammals, five PADI isozymes, modulate diverse cell and tissue functions, including gene expression, chromatin compaction, metabolism, nerve myelination, skin homoeostasis, the innate immune response, fertility and stem cell function (reviewed in ref. 2).

PADI4 possesses a nuclear localisation sequence and targets a variety of nuclear substrates, including core and linker histones[3–6], through which it modulates gene expression and chromatin compaction. Additionally, PADI4 modulates the innate immune response to

[1]Protein-Protein Interaction Laboratory, The Francis Crick Institute, London NW1 1AT, UK. [2]Epigenetics, The Babraham Institute, Cambridge CB22 3AT, UK. [3]MRC Human Genetics Unit, The University of Edinburgh, Western General Hospital, Edinburgh EH4 2XU, UK. [4]Antimicrobial Defense Laboratory, The Francis Crick Institute, London NW1 1AT, UK. [5]Structural Biology, The Francis Crick Institute, London NW1 1AT, UK. [6]Chemical Biology, The Francis Crick Institute, London NW1 1AT, UK. [7]Imaging, The Babraham Institute, Cambridge CB22 3AT, UK. [8]Proteomics, The Babraham Institute, Cambridge CB22 3AT, UK. [9]The University of Tokyo, Hongo, Bunkyo-ku, Tokyo 113-0033, Japan. [10]Imperial College London, Department of Chemistry, London W12 0BZ, UK. [11]These authors contributed equally: M. Teresa Bertran, Robert Walmsley. ✉e-mail: maria.christophorou@babraham.ac.uk; l.walport@imperial.ac.uk

infection and pro-inflammatory signalling and is strongly associated with neutrophil extracellular trap (NET) formation[7,8]. More recent evidence implicates PADI4 in the regulation of cell reprogramming[9].

Despite their wide range of physiological functions, the molecular mechanisms through which PADIs are regulated within cells are unknown. PADIs require calcium for catalysis, and the sequential binding of calcium ions results in a large structural rearrangement that forms the active site cleft[10,11]. However, the calcium concentrations required for activation in vitro (>1 mM), far exceed physiological intracellular concentrations (~100 nM)[12,13]. Whether cellular PADI activation results from a localised influx of calcium, or whether allosteric regulation mechanisms render PADIs capable of catalysis at intracellular calcium levels is not known.

PADIs are perhaps best known for their role in disease development: aberrant citrullination is associated with a host of pathologies, including autoimmunity (rheumatoid arthritis[14], ulcerative colitis[15], lupus erythematosus[16–18], and psoriasis[19]), neurodegeneration[20], age-associated tissue fibrosis[21], and cancer[22–24]. *Padi4* is one of the top susceptibility loci for rheumatoid arthritis (RA) and autoantibodies against endogenous citrullinated proteins serve as RA biomarkers[25]. Pharmacological or genetic perturbation of PADI4 in mice alleviates the development of RA and ulcerative colitis, while *Padi4-null* mice are protected against NET-associated tissue destruction in a wide variety of contexts[15,26]. There is thus overwhelming evidence for the therapeutic potential of PADI4 inhibition. Consequently, a range of PADI4 inhibitors have been developed in recent years, the majority of which are based on covalent substrate mimics[27,28]. However, due to high conservation between family members, it has been difficult to achieve good isozyme selectivity[29,30]. A recently developed non-covalent inhibitor of PADI4, GSK484[31], is significantly more potent and specific than other compounds and preferentially binds and stabilises the calcium-free form of the enzyme, inhibiting activation. GSK484 derivatives such as BMS-P5 and JBI-589 have been shown to inhibit NETosis and neutrophil chemotaxis[32–35]. However, despite this substantial progress, we still lack compounds of sufficient potency and specificity for clinical use. In settings of aberrantly high PADI4 activity, a reversible inhibitor that is potent and selective against the calcium-bound, active form of PADI is likely to be beneficial.

Beyond the obvious need to develop PADI inhibitors, pharmacological activators are likely to revolutionise our understanding of PADI regulation and function. Currently, stimuli such as calcium ionophores or lipopolysaccharide (LPS) are used to stimulate PADI activity in cell-based assays. These pleiotropic stimuli engage a wide range of biological pathways, making it impossible to disentangle the specific mechanistic contributions of PADIs. Specific and selective PADI activators would provide powerful tools for studying PADI-mediated cellular functions and may be used to enhance cellular manipulations such as induced cell reprogramming[9]. Importantly, such compounds can also inform our mechanistic understanding of PADI regulation, which may enable the rational design of selective PADI inhibitors that

block activation. The anti-parasitic and anti-viral compound nitazoxanide (NTZ) and the natural product demethoxycurcumin (DMC) have recently been identified as activators of PADI2[36,37]. Anti-PADI4 auto-antibodies found in RA patients have also been reported to activate PADI4, suggesting it may also be possible to activate PADI4 pharmacologically[38].

A wide range of chemical modalities have been explored for the modulation of protein targets, including small molecules, peptides and antibodies. Due to their intermediate size and chemical diversity, cyclic peptides are a powerful modality for developing modulators with exquisite potency and target selectivity. Such peptides can be identified from enormous libraries of DNA-encoded peptides using approaches such as phage display, mRNA-display, and its derivative RaPID that allows incorporation of non-natural amino acids into the peptide libraries[39–41].

Here, we tailored the conditions used in RaPID peptide selections to identify cyclic peptides with different biological activities, including both inhibitory and activating peptides, alongside neutral binders. To quantify PADI4 activity in cells, we identified PADI4-activating cellular stimuli and developed a high-content microscopy-based method. The lead inhibitory peptide, PADI4_**3**, binds the active form of PADI4 and inhibits activity both in vitro and in cells, with higher potency and isozyme selectivity than previous compounds. The lead-activating peptide, PADI4_**11**, lowers the calcium requirement for activation in vitro and induces activation within cells in the absence of any additional stimuli. Structural and mutagenesis studies reveal that PADI4_**11** activates PADI4 through allosteric binding to a region of PADI4 that rearranges substantially upon calcium binding to form the catalytically competent enzyme. Finally, we functionalised PADI4_**7**, an inert PADI4 binder, to generate a biotinylated probe, bio-PADI4_**7**. We use bio-PADI4_**7** to pull down PADI4 from cells under resting or activating conditions and report the proteins associated with PADI4 in these two contexts. Overall, this work provides an approach for the screening and optimisation of modulators against cellular PADIs, a comprehensive set of tools for the study of cellular PADI4 and a starting point for the development of PADI4-targeting therapeutics.

## Results
### A strategy for identifying potent and conformation-selective PADI4 binders

We set out to identify cyclic peptides that bound to different conformations of PADI4 and might, therefore, have different effects on its activity. To this end, we performed RaPID selections against recombinant human PADI4 (thereafter labelled PADI4) under three different conditions. To identify peptides that bound preferentially to either the calcium-bound "active" or calcium-free "inactive" form of the enzyme, selections were carried out using buffer containing concentrations of calcium (10 mM) known to achieve full PADI4 activity in vitro (Selection 1), or a calcium-free buffer (Selection 2) (Fig. 1). For the final selection we reasoned that mimicking the active, substrate-bound

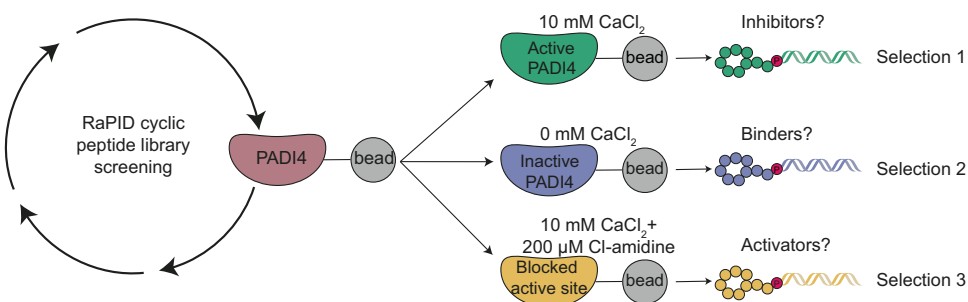

**Fig. 1 | Schematic of the RaPID selection strategies used to identify PADI4-binding cyclic peptides.** RaPID selections were carried out against PADI4 as bait under three different conditions to target different conformations of PADI4.

**Table 1 | Sequence and binding affinities against Bio-hPADI4 wt of solid-phase synthesised cyclic peptides identified by RaPID screening determined by surface plasmon resonance (SPR)**

| Selection conditions | Name | Sequence | Length | $K_D$ (nM) | | |
|---|---|---|---|---|---|---|
| | | | | 10 mM CaCl$_2$ | 0 mM CaCl$_2$ | 10 mM CaCl$_2$ + Cl-amidine |
| 10 mM CaCl$_2$ | **PADI4_1** | Cyclic-$_D$YFYRIGFWYPNYQC(S-)G-NH$_2$ | 15mer | 205 ± 52 | 105 ± 16 | 727 ± 174 |
| | **PADI4_2** | Cyclic-$_D$YRDHRSPFDGYC(S-)G-NH$_2$ | 13mer | 13 ± 4 | >5 μM | >5 μM |
| | **PADI4_3** | Cyclic-$_D$YRDHHYRHPKYC(S-)G-NH$_2$ | 13mer | 2.7 ± 0.5 | >5 μM | >5 μM |
| | **PADI4_4** | Cyclic-$_D$YHRLIVVIYVC(S-)G-NH$_2$ | 12mer | >2 μM | >2 μM | >2 μM |
| | **PADI4_5** | Cyclic-$_D$YATPWLIVVLC(S-)G-NH$_2$ | 12mer | >2 μM | >2 μM | >2 μM |
| | **PADI4_6** | Cyclic-$_D$YLVLTIRLVLC(S-)G-NH$_2$ | 12mer | >2 μM | >2 μM | >2 μM |
| 0 mM CaCl$_2$ | **PADI4_7** | Cyclic-$_D$YYPKGSWGYKLFC(S-)G-NH$_2$ | 14mer | 39 ± 24 | 8 ± 1 | 36 ± 12 |
| | **PADI4_8** | Cyclic-$_D$YTLWTVLVVIC(S-)CG-NH$_2$ | 12mer | >2 μM | >2 μM | >2 μM |
| | **PADI4_9** | Cyclic-$_D$YAQWYIWVLLC(S-)G-NH$_2$ | 12mer | >2 μM | >2 μM | >2 μM |
| | **PADI4_10** | Cyclic-$_D$YPWEISVWLLYC(S-)G-NH$_2$ | 13mer | >2 μM | >2 μM | >2 μM |
| 10 mM CaCl$_2$ + Cl-amidine | **PADI4_11** | Cyclic-$_L$YESC(S-)RYRQVLQL-NH$_2$ | 12mer | 457 ± 109 | >5 μM | 382 ± 27 |
| | **PADI4_12** | Cyclic-$_L$YEPC(S-)RFREILDL-NH$_2$ | 12mer | 678 ± 82 | >5 μM | 674 ± 191 |

All peptides are cyclised via a thioether bond between the N-terminus and the cysteine side chain.

form of PADI4 would select for peptides that stabilised this protein conformation and that might consequently activate PADI4 at lower calcium concentrations. This selection was therefore performed both in the presence of 10 mM CaCl$_2$ and with the active site occupied by the covalent pan-PADI inhibitor Cl-amidine (200 μM Cl-amidine) (Selection 3)[30]. We used in vitro translation of puromycin-ligated mRNA templates to produce libraries of cyclic peptides conjugated to a hybrid of the encoding mRNA/cDNA (>10$^{12}$ library members). Peptide sequences consisted of a random region of 6–12 amino acids flanked by an *N*-chloroacetyl-$_D$-tyrosine (Selection 1 and 2) or *N*-chloroacetyl-$_L$-tyrosine (Selection 3) initiator and a *C*-terminal cysteine to permit peptide cyclisation. Sheared salmon sperm DNA was included in the selection buffers to reduce high levels of non-specific nucleic acid binding to PADI4, which were observed during initial optimisations[42]. Following incubation of the encoded peptide library with bead-immobilised PADI4, non-binding library members were washed away. Peptide-nucleic acid hybrids bound to the bait were then recovered and amplified by PCR for use as the input library for a repeat round of selection. In total, eight or nine sequential rounds of selection were performed, and library DNA recovered from later rounds was sequenced to identify the most enriched sequences (Supplementary Data 1 and Fig. S1). Following sequence deconvolution, twelve of the most enriched peptide sequences were selected for solid-phase peptide synthesis (SPPS) and further characterisation (Table 1 and Table S1).

To validate the enriched sequences, we initially measured their binding affinities to PADI4 using surface plasmon resonance (SPR). Affinities were measured both in the presence and absence of calcium (Table 1 and Table S1). We identified two peptides that bound in the low nanomolar range both in the presence and absence of calcium (PADI4_1 and PADI4_7). Four others bound in the low-to-mid nanomolar range to the calcium-bound form of the enzyme but were unable to bind in the absence of calcium (PADI4_2, PADI4_3, PADI4_11 and PADI4_12). Of note, an unusually high proportion of peptides were highly enriched during the RaPID selection but did not appear to bind to PADI4 by SPR (PADI4_5, 6, 8, 9, 10, $K_D$ > 2 μM). We attribute this to the high background levels of nucleic acid binding, which were only partially reduced in the selections by including sheared salmon sperm DNA in the selection buffer. This may have resulted in the retention of peptides that bound primarily through their attached mRNA/DNA hybrid tag. These peptides were removed from subsequent analyses.

To further interrogate the mode of binding, we carried out SPR experiments under the conditions used in Selection 3 (10 mM Ca$^{2+}$,

PADI4 active site blocked with Cl-amidine). As expected, the two peptides identified from Selection 3 (PADI4_11 and PADI4_12) were unaffected by Cl-amidine occupying the active site (Table 1, Table S1). The peptides that bound to both the calcium-bound and calcium-free form of the enzyme (PADI4_1 and PADI4_7) were also unaffected. By contrast, the two peptides that bound in the low nanomolar range to calcium-bound PADI4 specifically (PADI4_2 and PADI4_3) were unable to bind once the active site was blocked by Cl-amidine. This indicated that PADI4_2 and PADI4_3 likely bind at the active site, whilst the others must bind elsewhere on the enzyme.

## PADI4_3 is a potent and selective cyclic peptide inhibitor of PADI4

Having identified several tight-binding cyclic peptides we next asked whether they were able to modulate PADI4 activity in vitro. IC$_{50}$ values were determined using an established colorimetric activity assay, the Colour Developing Reagent (COLDER) assay, and the model substrate $N^\alpha$-benzoyl-$_L$-arginine ethyl ester (BAEE)[43]. Consistent with the binding experiments described above, PADI4_2 and PADI4_3 both inhibited PADI4 activity with IC$_{50}$s of 209 ± 54 nM and 56 ± 7 nM, respectively (Fig. 2A). Note that as the IC$_{50}$ value for PADI4_3 is similar to the concentration of recombinant PADI4 enzyme used in the assay (50 nM) it is likely a slight underestimate[44]. No PADI4 inhibition was observed for the four peptides that were unaffected by blocking of the active site with Cl-amidine (PADI4_1, PADI4_7, PADI4_11 and PADI4_12). Similar inhibition of histone 3 citrullination (H3Cit) was observed with PADI4_2 and PADI4_3 when activity assays were performed in cell lysates containing active cellular PADI4 (Fig. S2A). In these assays, we also observed that inhibition was achieved without pre-incubating PADI4 with peptide prior to activating the enzyme with calcium, in contrast to the most potent and specific reported PADI4 inhibitor, GSK484, which required pre-incubation to be active[31].

Many inhibitors described to date (e.g. Cl-amidine) inhibit PADI family members non-selectively due to the high sequence and structural similarity between PADI isozymes, especially around the active site (Figure S2B)[29,30]. Given the propensity of cyclic peptides to exhibit high selectivity[45,46], we reasoned that our lead peptide, PADI4_3, might be selective for PADI4. To test this, we performed in vitro inhibition assays against the other active members of the PADI family, PADI1-3, and mouse PADI4. No inhibition was observed with up to 100 μM peptide against any other PADIs (Fig. S2C, D). This suggests PADI4_3 is more than 1700-fold selective for human PADI4 over other family

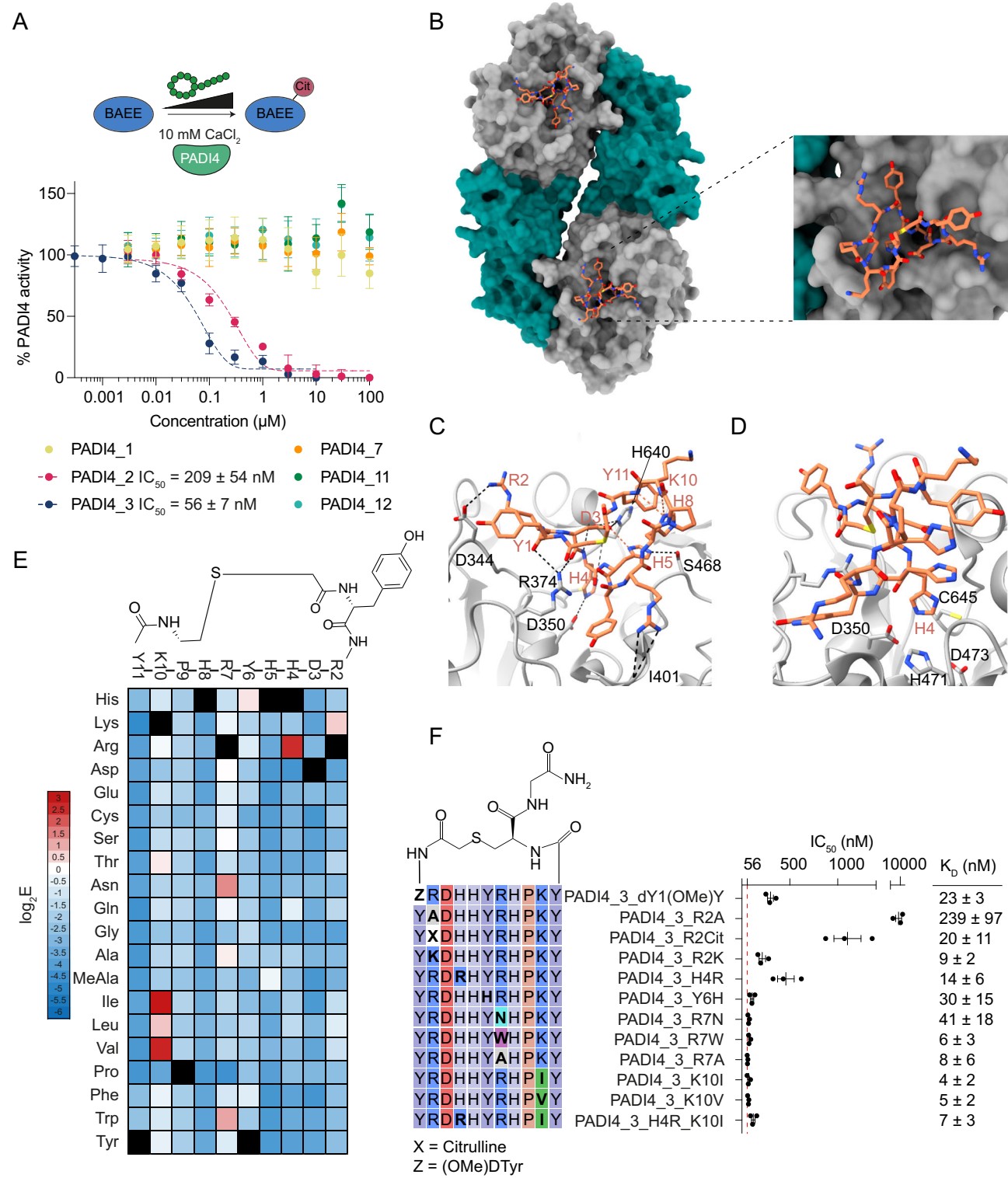

members, placing it among the most selective PADI inhibitors identified to date[47].

## Structural studies reveal that PADI4_3 inhibits PADI4 through active site binding

Having identified PADI4_3 as a potent and selective inhibitor of PADI4, we aimed to characterise its mode of binding. We determined the structure of PADI4_3-bound PADI4 by cryoEM to an overall resolution of 3.1 Å (Fig. 2B, Table S2 and Fig. S3A). The protein adopted a similar conformation to previously reported structures (e.g. PDB ID: 1WD9, all-atom rmsd: 0.57 Å) with PADI4 adopting a head-to-tail homodimeric structure. Additionally, well-defined density for the whole cyclic peptide was observed in both monomers, allowing us to assign all residues unambiguously (Fig. S3B). Unusually for peptides derived from RaPID selections, PADI4_3 itself does not adopt any secondary structure elements and relatively few intramolecular hydrogen bonds are observed within PADI4_3[48]. Those that are observed, such as between PADI4_3 H8 and K10 and Y11 and between PADI4_3 H5 and D3, are mainly side chain-to-backbone rather than backbone-to-backbone interactions (Fig. 2C).

**Fig. 2 | PADI4_3 is a potent inhibitor of calcium-bound PADI4. A** Inhibition of PADI4 by cyclic peptides identified in the RaPID selections. COLDER assays were performed at different peptide concentrations (10–0.0003 μM) in the presence of 10 mM CaCl$_2$. Data were normalised to activity of PADI4 in the presence of 0.1% DMSO. Data shows mean ± SEM of three independent replicates. Each replicate was done in triplicate. **B** Overall cryoEM structure of PADI4 homodimer (*N*-terminal immunoglobulin-like domains shown as grey surface, *C*-terminal catalytic domain as green surface) bound to PADI4_3 (orange sticks). Inset shows a zoomed-in view of the peptide binding site. **C** View from a cryoEM structure of PADI4 homodimer (grey cartoon) bound to PADI4_3 (orange sticks) highlighting a network of inter- and intramolecular interactions formed by PADI4_3. Intramolecular hydrogen bonds are shown as orange dashed lines, and intermolecular bonds as black dashed lines. **D** View from a cryoEM structure of PADI4 homodimer (grey cartoon) bound to PADI4_3 (orange sticks) highlighting the core catalytic residues of PADI4. His4 of PADI4_3 occupies the site bound by arginine residues in substrate peptides. **E.** Mutational scanning reveals which regions of PADI4_3 appear tolerant to amino acid substitutions. Columns show each position in the PADI4_3 sequence, and rows show individual amino acid substitutions. Data were plotted as the mean enrichment score from three independent selections where red represents amino acids with high binding affinity and blue represents amino acids with low binding affinity. Black squares highlight the parent amino acids. **F.** Sequence of the PADI4_3 analogues synthesised by SPPS (left) and their calculated IC$_{50}$ and $K_d$ values (right). Activity data was measured as in A and $K_d$ values determined by SPR. Both IC$_{50}$ and $K_d$ values show mean ± SEM from three independent replicates.

The structure confirmed that PADI4_3 is a substrate-competitive inhibitor that binds within the active site of PADI4. Although PADI4_3 occupies the substrate binding site, rather than directly mimicking substrate binding by directing an arginine towards the active site, a histidine residue, H4, occupies this position (Fig. 2D and Fig. S3C). 760 Å$^2$ of surface area is buried (per monomer) and, in line with its nanomolar binding affinity, there are many specific intermolecular interactions between PADI4 and PADI4_3 across this interface. This includes a bifurcated hydrogen bond between PADI4 R374 and the carbonyls of the PADI4_3 D3 backbone and the N-terminal acetyl group. Hydrogen bonds between the side chains of PADI4_3 R2 and PADI4 D344, PADI4_3 D3 and PADI4 H640, PADI4_3 H4 and PADI4 D350, and PADI4_3 H5 and PADI4 S468 are also observed (Fig. 2C). In line with the selectivity observed in COLDER assays, many of these residues are not conserved in other PADI isozymes or mouse PADI4. This is most pronounced for H640, which is not conserved in PADI1, PADI2 or mPADI4, and D344, which is not conserved in any of the other human PADIs (Fig. S3D–F).

In parallel, we applied a deep mutational scanning approach reported by Rogers et al. to provide an orthogonal view of which PADI4_3 residues were important for binding. We also hoped to understand whether substitutions might improve potency[49]. A scanning library was designed and synthesised based on the PADI4_3 peptide sequence, in which positions 2–11 in the peptide were individually randomised to every other coded amino acid (except methionine, which was reprogrammed as *N*-methyl-alanine). A single round of selection was performed by panning this displayed library against immobilised PADI4 under conditions in which equilibrium library binding was achieved in each of the three wash steps. Next-generation sequencing of the input and bound library was performed and an enrichment score (E) for each peptide was calculated based on the ratio of the frequency of the peptide in the bound library relative to the input library. A negative log$_2$E value is suggestive of a binder that is weaker than the PADI4_3 parent sequence, whilst a positive value indicates tighter binding.

Our analysis of the high-throughput sequencing data suggested that the amino acids in the parent sequence were the optimal binders in many positions (Fig. 2E and Fig. S4). However, we did observe positive enrichment scores for some modified peptide sequences, indicating potentially favourable amino acid substitutions e.g. R7N (E = 1.37 ± 0.25) and K10I (E = 2.8 ± 0.49). These observations correlated well with our cryoEM structure; largely solvent-exposed residues in the cryoEM structure, such as P9 and K10, tended to be more tolerant to substitutions (indicated by most amino acid substitutions having only slightly negative E values), whilst those where we observed specific contacts with the protein, such as residues R2-H5, had highly negative enrichment scores for almost all substitutions apart from those most similar to the amino acid in the parent sequence.

To test whether these observations correlated with changes in binding affinity, we synthesised and characterised a series of peptide variants that either showed positive log$_2$E values or that our structural data suggested would abrogate binding. While most of the peptide variants with positive log$_2$E values showed similar binding affinities and IC$_{50}$s to the parent sequence, surprisingly, none were more potent than PADI4_3 (Fig. 2F, Fig. S5 and Table S2). Nonetheless, several positions showed significant flexibility, such as R7, which tolerated amino acids with very different chemical properties in this position, including tryptophan, asparagine and alanine (Fig. 2F, Fig. S5 and Table S2). By contrast, our structure suggested that R2 interacts with D344 on the protein (Fig. 2C), and scanning data suggested that a lysine residue was the only other natural amino acid permitted in this position, suggesting that positive charge is important in position 2. Consistent with this, changing R2 to alanine had a significant negative effect on binding and inhibition (PADI4_3_R2A: $K_D$ = 240 ± 97 nM, IC$_{50}$ = 9.4 ± 2.1 μM), whilst substitution with lysine substantially recovered binding and inhibition (PADI4_3_R2K: $K_D$ = 9 ± 2 nM, IC$_{50}$ = 213 ± 59 nM). Introducing citrulline at this position also partially recovered activity (PADI4_3_R2Cit: $K_D$ = 20 ± 11 nM, IC$_{50}$ = 1.1 ± 0.24 μM), suggesting that positive charge, though important, is not essential (Fig. 2F, Fig. S5 and Table S2). Finally, changing H4 for arginine was one of the most permitted substitutions in our mutational scanning experiment; however, substitution of H4 for arginine had a negative impact on the IC50 of the peptide (PADI4_3_H4R: $K_D$ = 14 ± 6 nM, IC$_{50}$ = 450 ± 149 nM). As PADI4_3 H4 sits in the catalytic pocket of PADI4, in the position usually occupied by substrate arginine residues (Figure S3C), we wondered whether this poorer correlation was due to this peptide becoming a substrate of PADI4. To test this, we performed COLDER assays comparing PADI4_3_H4R with a model substrate and concluded that peptide PADI4_3_H4R is not a substrate of PADI4 (Fig. S6).

## Development of a high-content imaging method for the quantification of cellular PADI4 activity

We went on to test whether PADI4_3 can inhibit cellular PADI4. To enable comparative and quantitative measurements of peptide activity in cells, we set up a high-content microscopy method based upon the quantification of cellular citrullinated histone H3 (H3Cit) levels, as a read-out of cellular PADI4 activity, at single-cell resolution by immunofluorescence. To allow for tractable measurements, we aimed to identify a non-transformed, adherent cell type and a stimulus that can achieve PADI4 activation in the absence of calcium ionophores, which can be damaging to cells. Based on previous findings that PADI4 promotes embryonic stem cell pluripotency[50], we hypothesised that it may be catalytically activated under cell culture conditions that promote the naïve pluripotency state. This can be achieved by switching from primed pluripotency culture conditions (provision of 10% foetal bovine serum (FBS), and leukaemia inhibitory factor (LIF), hereafter referred to as "Serum" medium) to naïve pluripotency culture conditions, which involves provision of 10% Knock-out Serum Replacement (KSR), 1% FBS, LIF and inhibitors to kinases MEK and GSK3 (2i) (hereafter referred to as "KSR/2i" medium). As PADI4_3 is highly specific for human PADI4 and shows no efficacy against mouse PADI4 (Fig. S2D), for these assays, we generated mES cells stably expressing human

PADI4 (hPADI4-stable). Quantification of mean H3Cit intensity in these cells shows that transition from Serum to KSR/2i medium for 3 h is sufficient to induce a measurable induction of intracellular PADI4 activity (Fig. S7A). This method is quantitative, as demonstrated by the dose-response read-outs achieved by treatment with increasing doses of Cl-amidine (Fig. S7B).

## PADI4_3 is a cell-active PADI4 inhibitor

To achieve accurate downstream characterisation of PADI4_3 function, we generated a matched inactive control peptide where the second and third amino acids of the sequence were reversed (PADI4_3i). This negative control peptide did not bind or inhibit PADI4 (Fig. S8A, B) and was used in parallel with the parental PADI4_3 peptide in cell-based assays described below. To assess the cell permeability of these peptides, we used the established chloroalkane penetration assay (CAPA) cell penetration assay[51]. CAPA assays performed with PADI4_3 and PADI4_3i appended with chloroalkane groups showed that both peptides entered cells, with $CP_{50}$ values of $1.4 \pm 0.1\,\mu M$ and $2.0 \pm 0.8\,\mu M$, respectively (Fig. S8C) and that PADI4_3 enters the cells through active transport mechanisms ($CP_{50}$ at 4 °C of $9.7 \pm 2.1\,\mu M$) (Fig. S8D).

We then employed the high-content imaging method described above to quantify the efficacy of cyclic peptides against cellular PADI4. hPADI4-stable mES cells were stimulated with KSR/2i medium for 3 h, in the absence or presence of a range of PADI4_3 or PADI4_3i peptide concentrations. We found that PADI4_3 inhibits intracellular PADI4 in a dose-dependent manner (Fig. 3A–C). PADI4_3 inhibited PADI4 activity with an $EC_{50}$ of $0.2\,\mu M$. Treatment with $1\,\mu M$ was sufficient to achieve complete inhibition, while treatment with PADI4_3i resulted in no inhibition (Fig. 3B, C). No cytotoxicity was observed in the treatment of cells with PADI4_3 (Fig. S8E).

PADI4 activation is associated with the formation and function of neutrophil extracellular traps (NETs), web-like structures of decondensed extracellular chromatin decorated with antimicrobial proteins. NETs are released by neutrophils via a programmed cell death called NETosis as part of the innate immune response to infection and act as a defence mechanism for the capture and killing of diverse pathogens. However, NETs also occur during sterile inflammation and are associated with tissue destruction[52]. A range of reports demonstrate that genetic or pharmacological inhibition of PADI4 results in decreased extracellular histone citrullination and is protective against NET-associated tissue damage in a variety of pathological contexts, from rheumatoid arthritis and diabetes, to cancer metastasis and age-related tissue fibrosis[21,26,53–55]. It is, therefore, highly likely that inhibition of PADI4 in clinical settings will be beneficial. To test whether PADI4_3 can inhibit NET-associated citrullination in a physiologically relevant system, we assessed its effect on primary human peripheral neutrophils from healthy donors, treated with cholesterol crystals, a well-established inducer of NETosis[56]. We found that citrullination of histone 3 in NET preparations was abolished by PADI4_3 but not PADI4_3i (Fig. 3D).

The lack of citrullination resulting from PADI4 inhibition is often interpreted as a disruption in NET formation[8,31,53–55]. Conflicting findings, often arising from highly diverse and pleiotropic stimuli used to induce NETosis in different studies, have rendered it unclear whether PADI4 activation is a necessary step in the execution of NETosis[57]. We exploited the high potency and specificity of PADI4_3 and the availability of its negative control peptide, PADI4_3i, to assess the role of PADI4 in NETosis induced by different stimuli in primary human peripheral neutrophils. Inhibition of PADI4 with PADI4_3 ($50\,\mu M$) reduced H3Cit levels and NETosis in human neutrophils treated with calcium ionophore, while neither response was affected by incubation with the control peptide, PADI4_3i ($50\,\mu M$) (Fig. 3E–G). Other NETosis inducers, such as phorbol myristate acetate (PMA) or cholesterol crystals, have been reported to induce NETosis in a PADI4-independent manner[56,58,59]. Consistent with this, no reduction in NETosis was observed by treatment with PADI4_3

following induction with PMA or cholesterol crystals (Fig. S9), despite the fact that citrullination was abolished (Fig. 3D). Together, these results confirmed that the requirement for PADI4 activity in the activation of NETosis in human neutrophils is stimulus-dependent. The varying effects of PADI4_3 on NETosis elicited by different stimuli lends support to a mechanism by which inhibition of citrullination is achieved by direct engagement between PADI4_3 and PADI4, rather than a non-specific effect on NETosis, or by cell toxicity.

## PADI4_11 is a cell-permeable PADI4 activator

We hypothesised that the peptides that bound away from the active site may activate PADI4. To test this, we performed COLDER activity assays, titrating the calcium concentration at a constant concentration of cyclic peptide ($30\,\mu M$). In the absence of peptide, PADI4 required $225 \pm 10\,\mu M$ calcium to achieve 50% of maximal activity under saturating calcium concentrations ($K_{50(Ca2+)}$)[60]. Whilst little change in $K_{50(Ca2+)}$ was observed in the presence of PADI4_1 and PADI4_7, the calcium requirement was reduced 4-fold with PADI4_11 and PADI4_12, the two peptides identified from selection 3 (Fig. 4A). Additionally, even under saturating calcium concentrations, the presence of PADI4_11 or PADI4_12 increased PADI4 activity $36 \pm 9\%$ ($p < 0.0001$) or $12 \pm 15\%$ ($p = 0.03$) over the dimethylsulfoxide (DMSO) vehicle control respectively, suggesting a hyper-active conformation of PADI4 in the presence of either of these two peptides. This activation was dependent on the peptide concentration, requiring $1.6 \pm 0.6\,\mu M$ PAD4_11 and $6.5 \pm 0.1\,\mu M$ PAD4_12, to achieve 50% activation ($AC_{50}$) at 0.1 mM calcium (Fig. 4B). As for the inhibitory peptides, we also confirmed the isozyme specificity of both PADI4_11 and PADI4_12 using COLDER activity assays to test all the catalytically active human PADIs and mPADI4 (Fig. S10A–D). At $30\,\mu M$ peptide, no activation was observed for any other PADI. We also tested whether PADI4_11 was a substrate of PADI4 (Fig. S10E). A small, but statistically significant, increase in citrullination activity was observed in the presence of PADI4_11 ($400\,\mu M$) relative to the no substrate control. This may suggest that in addition to acting as an activator, PADI4_11 is also a very poor substrate of PADI4, although the small increase in citrullination activity observed may also be due to activation by PADI4_11 allosteric binding enhancing the amount of PADI4 autocitrullination observed[61–63].

Given that PADI4_11 was the most potent, we took it forward into cell-based assays. We synthesised a negative control, PADI4_11i, which consists of the parental sequence with the second and third amino acids reversed. This showed no binding to PADI4 ($K_d > 5\,\mu M$) (Fig. S11). We again assessed the cell permeability of both peptides using the CAPA assay. This demonstrated that both PADI4_11 and PADI4_11i are cell-permeable, with $CP_{50}$ values of $3.1 \pm 1.2\,\mu M$ and $2.0 \pm 0.5\,\mu M$, respectively (Fig. S10F), and PADI4_11 enters the cell through active transport ($CP_{50}$ at 4 °C is $10.8 \pm 4.7\,\mu M$) (Fig. S10G). Using our high-content imaging method for quantification of cellular H3Cit, we found that PADI4_11 can induce PADI4 activation in a dose-dependent manner in PADI4-stable mES cells without any other activating stimulus, while PADI4_11i does not have the same effect (Fig. 4C–E). Treatment of wild-type mES cells (which do not harbour human PADI4) with PAD4_11 did not induce citrullination at any concentration tested (Fig. 4F). Since calcium influx resulting from a compromise to cell membrane integrity might be sufficient to induce intracellular PADI4 activation, we tested whether PADI4_11 results in increased calcium influx, using live cell calcium imaging. We found that neither PADI4_11, nor PADI4_11i, were responsible for increased intracellular calcium levels (Fig. 4G and Fig. S12A). Similarly, PADI4_11 did not compromise cell membrane integrity, as demonstrated by the fact that treatment with peptide did not make the cells permeable to propidium iodide (PI) (Fig. 4H). These results also indicated that PADI4_11 was not toxic to the cells. We confirmed this using live cell imaging of cell death over a period of 48 h after treatment (Fig. S12B). We therefore conclude that the activating effect of PADI4_11 is specific and on-target.

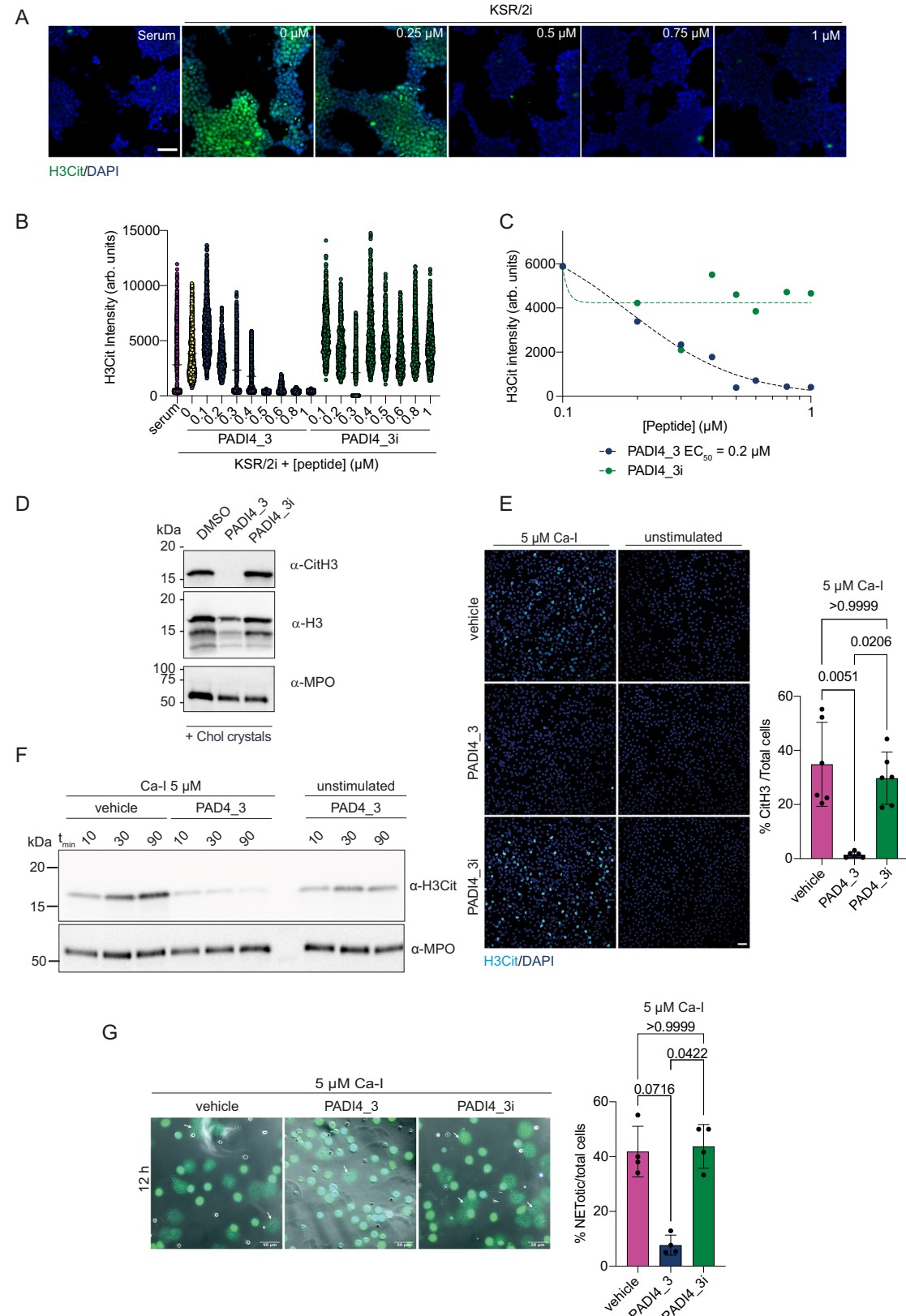

## The mechanism of action of PADI4_11 provides proof-of-principle for allosteric regulation of PADI4

As discussed above, it is currently unclear how PADI4 activation is achieved within cells at physiological calcium concentrations. One hypothesis is that a mode of allosteric regulation, such as a post-translational modification or binding to an interacting protein, elicits a conformational change that allows activation of the enzyme at lower calcium levels. Given that PADI4_11 can induce activation of PADI4 at fourfold lower calcium, we sought to understand the underlying mechanism. We first excluded direct calcium binding by the peptide itself through isothermal titration calorimetry between PADI4_11 and calcium (Fig. S13).

**Fig. 3 | PADI4_3 inhibits PADI4 in mES cells and human neutrophils.**
**A** Representative high-content immunofluorescence images of hPADI4-stable mES cells stimulated with KSR/2i in the presence of increasing concentrations of PADI4_3 for 3 h. Immunostaining of citrullinated histone H3 (H3Cit) is shown in green, and DAPI in blue. Scale bar = 50 µm. **B** High content imaging-based quantification of mean H3Cit immunofluorescence intensity in hPADI4-stable mES cells stimulated with KSR/2i in the presence of increasing concentrations of PADI4_3 or PADI4_3i for 3 h. Data represent mean ± SD from at least 500 cells from a representative experiment. **C** High content imaging-based quantification of mean H3Cit immunofluorescence intensity in hPADI4-stable mES cells treated with increasing concentrations of PADI4_3 or PADI4_3i for 3 h. Each data point represents the average H3Cit intensity per condition of one experiment. **D** Immunoblots showing citrullinated histone H3, total histone H3 and MPO levels from NET preparations following stimulation with 1 mg/ml cholesterol crystals and either DMSO (0.5%), 50 µM PADI4_3 or 50 µM PADI4_3i control. Data show a representative experiment after two repeats. **E** (Left) representative immunofluorescence confocal micrographs of human neutrophils pre-incubated with vehicle, PADI4_3 or PADI4_3i peptide for 60 minutes and stimulated with Ca-Ionophore (Time stamp = 12 h). Citrullinated H3 is shown in cyan and DAPI in blue. Scale bar = 40 µm. (right) Quantification of the immunofluorescence micrographs of the percentage of citrullinated H3-positive cells over the total number of cells. Data represent mean ± SD from six technical replicates. Kruskal–Wallis test was used to determine if the differences among means were significantly different from each other. p-values were adjusted using Dunn's correction for multiple comparisons. Data show a representative experiment after two repeats. **F** Cell lysates from calcium ionophore-stimulated and unstimulated human neutrophils pre-incubated with vehicle or PADI4_3 peptide (50 µM). Cells were collected at 10, 30 and 90 min post-stimulation and immunoblotted for citrullinated H3 and MPO. **G** (Left) Representative bright field micrographs of human neutrophils pre-incubated with vehicle, PADI4_3 or PADI4_3i peptide, imaged 12 h post-stimulation with Ca-ionophore. (Right) Quantification of the percentage of NETotic cells over total cells imaged in the micrographs. Quantification was done as in (**E**).

We then used cryoEM to elucidate the mode of binding of PADI4_11 to PADI4 and were able to obtain a structure with an overall resolution of 3.6 Å. As for the structure with the inhibitor PADI4_3 described above, PADI4 adopts a homodimeric structure. In this case, density was observed for all residues of the cyclic peptide except for the thioether cyclisation bridge (Fig. 5A, Table S2 and Fig. S14A). PADI4_11 adopts an alpha-helical structure with the internal cyclisation bridge between C4 and the N-terminus of the peptide acting like a peptide staple[64,65]. A characteristic pattern of intramolecular hydrogen bonds between backbone N-H groups and backbone carbonyl groups four residues preceding them stabilise this helical structure (Fig. 5B). Many intermolecular interactions are also observed between the peptide and PADI4, consistent with its tight-binding affinity (Fig. 5C and Fig. S14B).

The structure revealed that PADI4_11 binds to a surface of PADI4 distal to the active site, but close to the five calcium ion binding sites. The peptide binds in a cleft formed between two loops in PADI4 (D155-V171 and P371–387). These loops are flexible in the calcium-free form of PADI4 (no density is observed in the crystal structure of apo-PADI4 (PDB ID: 1WD8)) and rigidify upon calcium binding, in a structural rearrangement that forms the catalytically competent active site (Fig. S14C)[10,11]. PADI4_11 binding in this region thus fits well with our observations that PADI4_11 activates PADI4 by reducing its calcium dependency, since peptide binding likely alters the dynamics of this region and may, therefore, directly increase calcium binding affinities. In addition, and in line with the selectivity shown in the COLDER assays, these regions are not well conserved in other PADI isozymes (Fig. S14D–G).

We also performed an alanine scan of PADI4_11, which revealed that residues E2, R5 and R7 were particularly important for binding and activation (Fig. 5D). Substitution of R5 and R7 with alanine almost abrogated their binding to and activation of PADI4, which was restored by changing either residue to a lysine. This highlighted the necessity of a positively charged amino acid in these positions, potentially to interact with negatively charged amino acids in PADI4. Consistent with these observations, R7 makes direct interactions with PADI4 E167 in the cryoEM structure, though relevant interactions between PADI4 and R5 are less obvious (Fig. 5C). By contrast, substitution of E2 by alanine, aspartic acid or glutamine all resulted in loss of activity, suggesting the importance of the appropriate side chain length, as well as negative charge in position 2 of PADI4_11, in allowing it to form a salt bridge with R394 in PADI4 (Fig. 5C). Although no amino acid in the linear region of PADI4_11 (residues 8-12) appeared critical individually, a truncated 6mer peptide, PADI4_11b, was inactive (Fig. 5D). In addition, we could confirm that cyclisation is necessary for binding, as substitution of C4 to alanine abrogated binding and activity of the peptide.

To further verify the allosteric activation mechanism of PADI4_11 predicted by the structure, we produced several PADI4 variants, each with a single amino acid substitution, which we hypothesised might alter peptide binding (Table 2 and Fig. S15). To assess the activity of the different variants, we determined their $K_{50(Ca2+)}$ values. As described by Arita et al.[10], the substitution of N373 for an alanine completely abolished PADI4 activity. Substitution of all other amino acids tested had minimal effects on the maximal catalysis rate when saturated with calcium but, in most cases, increased their $K_{50(Ca2+)}$ values (up to 14-fold for PADI4_D165A).

We then went on to investigate how well PADI4_11 activated each of these variants. We determined the change in $K_{50(Ca2+)}$ for each protein variant on the addition of PADI4_11 (30 µM) and compared this with the change in $K_{50(Ca2+)}$ for the wild-type PADI4. We also measured the binding affinity of PADI4_11 to each variant. Incubation of the peptide with the catalytically inactive variant (N373A) did not rescue its activity. Consistent with our alanine scan data, the substitution of R394 (predicted to form a salt bridge with E2 in PADI4_11) with an alanine produced a PADI4 variant that did not bind to, and consequently was not activated by PADI4_11, confirming that R394 is essential for the activation mechanism.

Our final four variants spanned a flexible loop in PADI4 (D165-D168) that rearranges upon calcium binding and appeared from our cryoEM structure to be involved in PADI4_11 binding. The $K_{50(Ca2+)}$ value for variant D168A decreased by about threefold when incubated with PADI4_11, almost to the same degree as with the wild-type protein (Table 2 and Fig. S15). Variant E167A also displayed a similar fold change in $K_{50(Ca2+)}$ on PADI4_11 binding as the wild-type enzyme. However, the binding affinity of this variant to PADI4_11 was about 10-fold reduced, consistent with the observed interaction between PADI4 E167 and PADI4_11 R7. This data suggested that E167 and D168 are not essential for PADI4 activation by PADI4_11. By contrast, the $K_{50(Ca2+)}$ for PADI4_D165A changed very little when incubated with PADI4_11, suggesting D165 is critical for activation by PADI4_11. Substitution of C166 with bulky phenylalanine (as observed in the mouse PADI4 sequence) also completely abolished activation of the variant by PADI4_11. Together, these results reveal that the region from 165-168 is necessary for the activation of PADI4 by PADI4_11, although not all the residues in the loop are essential for this binding. Additionally, these findings point to a critical portion of PADI4, which may be involved in the allosteric regulation of the enzyme by other cellular enzymes or interacting proteins.

## A nanomolar, inert PADI4-binding peptide functionalised as a pull-down reagent

In addition to our inhibitory and activating peptides, there remained two potent binders, PAD4_1 and PAD4_7, which neither inhibited nor activated PADI4 catalytic activity in vitro (Figs. 2A, 4A). Cyclic peptides with similar affinities to other protein targets, have previously been functionalised as probes for a variety of assays[66,67]. We, therefore,

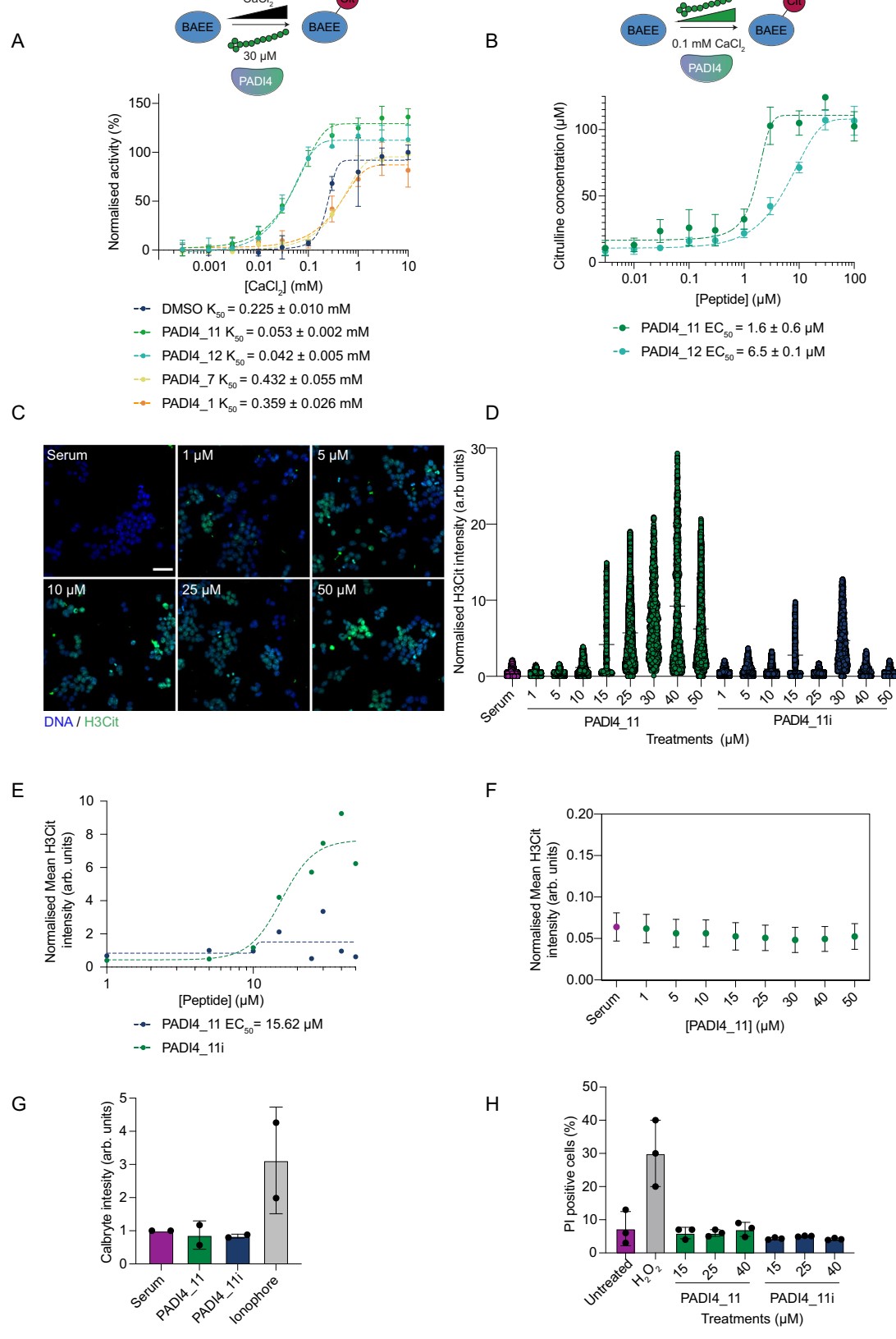

sought to generate a pull-down probe from PADI4_7 (the tighter binder of the two peptides). We synthesised a biotinylated variant (bio-PADI4_7) and tested its ability to enrich PADI4 from our PADI4-stable mES cells, or control cells which only harbour endogenous mouse PADI4. Immunoblot analysis of pull-down samples demonstrated significant enrichment when bio-PADI4_7 was used as the bait (Fig. 6A).

Thus, bio-PADI4_7 can be used to enrich cellular PADI4 and for subsequent biochemical analyses. To explore the binding partners of PADI4, we analysed pull-downs of these cells with bio-PADI4_7, or its non-binding scrambled variant bio-PADI4_7scr, by mass spectrometry. PADI4 is highly enriched in bio-PADI4_7 pull-downs as compared with bio-PADI4_7scr (Fig. 6B, C and Fig. S16A). In addition to PADI4, we

**Fig. 4 | PADI4_11 is a PADI4 activator. A** PADI4_11 and PADI4_12 are PADI4 activators in vitro. Activation COLDER assays with PADI4 and different PADI4-binding peptides. Assays were performed with different concentrations of CaCl$_2$ (10–0.003 mM) and 30 μM peptide. K$_{50Ca2+}$ is the concentration of CaCl$_2$ that yields half maximum PADI4 activity. Data were normalised against the activity of PADI4 in the presence of 0.1% DMSO and 10 mM CaCl$_2$ and represents mean ± SEM of three independent replicates. Each replicate was done in triplicate. **B** PADI4_11 and PADI4_12 activate PADI4 at low calcium concentrations. COLDER assays were performed at different concentrations of peptides (0.003–100 μM) and 0.1 mM CaCl$_2$. Data represents mean ± SEM of three independent replicates done in triplicate. **C** Representative high-content immunofluorescence images of hPADI4-stable mES cells in the presence of increasing concentrations of PADI4_11 for 1 h. Citrullinated histone H3 (H3Cit) is shown in green, and DAPI in blue. Scale bar = 50 μm. **D** High content imaging-based quantification of mean H3Cit immunofluorescence intensity in hPADI4-stable mES cells treated with increasing concentrations of PADI4_11 or PADI4_11i for 1 h. Each data point represents the mean H3Cit intensity per cell. All cells from three technical replicates and three individual experiments are included. **E** High content imaging-based quantification of mean H3Cit immunofluorescence intensity in hPADI4-stable mES cells treated with increasing concentrations of PADI4_11 or PADI4_11i for 1 h. Each data point represents the average H3Cit intensity per condition from three technical replicates and three individual experiments. **F** High content imaging-based quantification of mean H3Cit immunofluorescence intensity in control mES cells treated with increasing concentrations of PADI4_11 or PADI4_11i for 1 h. Each data point represents the mean ± SD H3Cit intensity of at least 3000 cells per condition from a representative experiment. **G** Calcium influx into cells, as measured by Calbryte intensity, after treatment with 25 μM PADI4_11 or PADI4_11i. Calcium ionophore A23187 (10 μM) was used as a positive control for calcium influx. Data represent mean ± SD of two biological replicates. **H** Flow cytometry-based quantification of propidium iodide (PI) incorporation into cells, as a measure of cell toxicity and membrane permeability, after treatment with increasing concentrations of PADI4_11 or PADI4_11i. Hydrogen peroxide (H$_2$O$_2$) is used as a positive control for cell toxicity. Data represent mean ± SD of three biological replicates.

observed a number of other proteins enriched by bio-PADI4_7 (Fig. 6B, C). These may include PADI4-interacting proteins, as confirmed for two of the candidates by pull-down followed by immunoblot (Fig. S16B). To obtain an understanding of the PADI4 interactome under PADI4 resting or activation conditions, we performed bio-PADI4_7 and bio-PADI4_7scr pull-downs from PADI4-stable mES cells stimulated with KSR-containing medium, or grown in fresh serum-containing medium, for 3 h. We observe a different set of PADI4-interacting proteins in Serum or KSR conditions (Fig. 6D). Given that the stimulation of the cells with KSR happened over a 3 h window, we do not expect the proteome of the cells to be different, suggesting PADI4 may associate with a different set of proteins under the two different conditions. The full proteomic dataset obtained from these experiments is provided in Supplementary Data 2.

## Discussion

Mounting evidence underscores the importance of PADI deregulation in disease, while emerging findings suggest that PADIs have wide ranging physiological roles. Chemical tools that allow researchers to finely tune or perturb the activity of single PADI isozymes will be invaluable in deciphering the roles of these enzymes in cell physiology and are likely to aid the development of important therapeutic and cell modulatory agents. In this work, we developed three classes of potent and isozyme-selective PADI4-binding cyclic peptides.

Firstly, we identified PADI4_3, a highly specific, reversible PADI4 inhibitor that exhibits high potency in cells. Whilst a range of PADI4 inhibitors have been reported previously, very few show a high degree of selectivity for a single PADI isozyme[47]. GSK484[31] is the most potent and selective PADI4 inhibitor reported to date. However, GSK484 binds preferentially to the calcium-free, inactive form of PADI4. In disease contexts, such as autoimmunity, cancer, atherosclerosis and fibrosis, where PADI4 expression or activity are aberrantly high, a potent and selective inhibitor, which is specific for the active form, is likely to be highly advantageous. PADI4_3, therefore, provides a starting point for the development of clinically relevant compounds with potential efficacy in a variety of pathological contexts.

Secondly, we identified PADI4_11, a synthetic activator of PADI4, that reduces the requirement of PADI4 for calcium in vitro and achieves specific activation of PADI4 in a cellular context. This reagent has the potential to revolutionise our understanding of the cellular consequences of PADI4 activation, which can only be achieved currently using highly pleiotropic stimuli. For example, such a reagent may be applied in contexts where PADI4 activation has a cell modulatory effect, such as stem cell and induced cell reprogramming[9], to increase our understanding of the signalling and epigenetic events that mediate changes in cell state. Additionally, as PADI4 activation has been shown to occur during, and to mediate cell reprogramming,

PADI4_11 may be developed as a tool to enhance and tightly control the reprogramming process.

Importantly, PADI4_11 also informs our molecular understanding of cellular PADI4 activation. The structural and mutagenesis analyses presented here reveal that PADI4_11 binds at an allosteric site on PADI4, in a region of the protein that is disordered in calcium-free crystal structures but becomes structured upon calcium binding[10,11]. Detailed structural analyses of calcium binding to PADI2 by Thompson and co-workers suggest that partial calcium binding (at sites 3, 4 and 5) reorients one of these loops (residues 369–389 in PADI2, corresponding to 368-388 in PADI4) forming the calcium 2 binding site, which once occupied by a calcium ion then allows catalysis. Thus, it is conceivable that, by binding in this same region, PADI4_11 promotes this same rearrangement, allowing the catalytically competent conformation of PADI4 to form at reduced calcium concentrations, either by increasing the calcium-binding affinity for one of the binding sites or by removing the requirement for calcium binding in one of these sites. It is worth noting that molecular docking studies suggest that the PADI2 activator DMZ potentiates PADI2 activity by binding at a similar site, though whether it reduces the requirement of PADI2 for calcium is unknown[37]. Darrah et al., also previously reported a class of anti-PADI4 autoantigens from rheumatoid arthritis patients that decreased the concentration of calcium required for PADI4 activity[38]. Proteolysis experiments suggested these antibodies bind at the same site as PADI4_11. Taken together, these findings point to potential modes of allosteric regulation in vivo, whereby PADI4 modifications or protein interactions, which may result from signalling elicited by inflammatory or developmental stimuli, may allow activation at physiological levels of intracellular calcium. Interestingly, recent studies have reported the binding of several intracellular proteins in this same region, both on PADI4 and PADI2[68,69]. Whilst no functional consequence on PADI activity has been reported for binding of these other proteins, combined with the known autoantibodies and our data, this raises the possibility that this region represents a hotspot for protein binding partners, or protein modifying enzymes, that activate PADIs. A recent study used antibody engineering to identify regulators of PADI4 enzymatic activity and provides additional evidence that PADI4 activation can be regulated through allosteric binding[70], strengthening the hypothesis that allosteric mechanisms may operate in vivo.

Finally, we report PADI4_7, an inert but potent PADI4 binder and demonstrate that a functionalised, biotinylated version, bio-PADI4_7, can isolate PADI4 from whole cell extracts. The high specificity and small size of this cyclic peptide reagent may eliminate the non-specific binding that is often unavoidable with the use of antibodies and may be developed further for additional applications, for example, for fluorescence-based PADI4 localisation studies. As our understanding of physiologically relevant contexts of PADI4 activation increases,

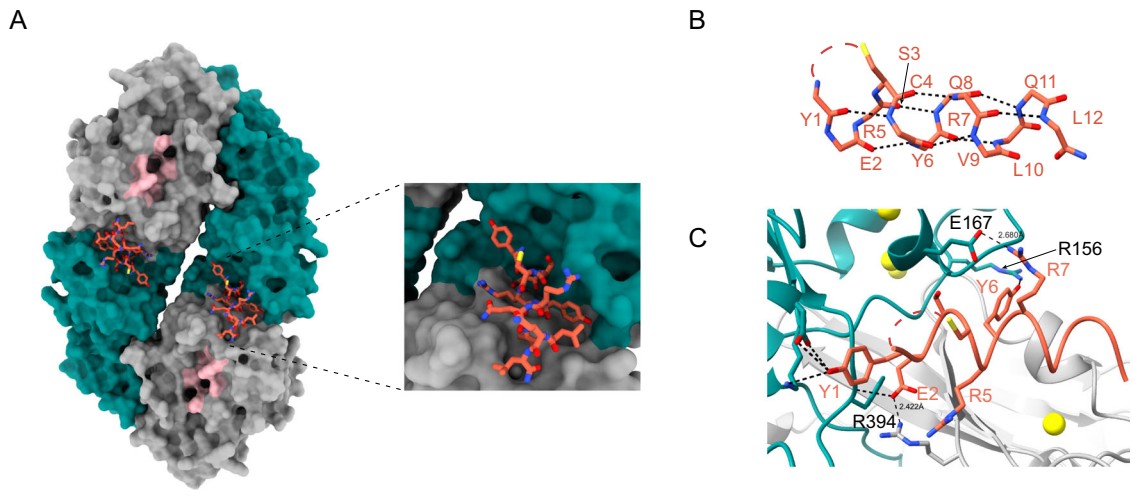

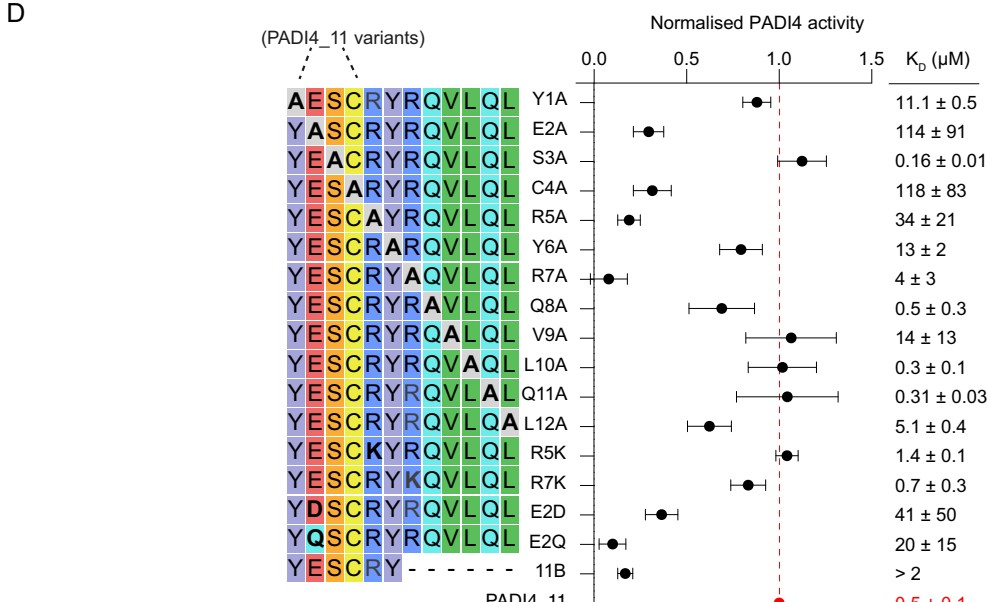

**Fig. 5 | PADI4_11 activates PADI4 by allosteric binding. A** Views from a cryoEM structure of PADI4 homodimer (*N*-terminal immunoglobulin-like domains in grey surface, *C*-terminal catalytic domain in green surface and active site in pink) bound to PADI4_11 (coral sticks). The inset shows a zoomed-in view of the peptide binding site. **B** View of PADI4_11 forming an alpha-helical structure. Intramolecular backbone interactions are shown as black dashed lines. **C** View from PADI4 homodimer (grey and cyan) bound to PADI4_11 (coral) showing a network of intramolecular interactions between PADI4_11 and PADI4. Interactions are shown as black dashed lines. **D** (Left) Sequence of the different PADI4_11 analogues synthesised by SPPS and in vitro activity measured by COLDER assay (centre). Assays were performed with 50 nM PADI4 in the presence of 0.1 mM CaCl$_2$ and 100 μM peptide. Data were normalised against the activity of PADI4 with 0.1 mM CaCl$_2$ and 100 μM of PADI4_11. $K_D$ values were measured by SPR (right). Data represents mean ± SEM of three independent repeats. Each COLDER experiment was performed in technical triplicate.

**Table 2 | Summary of binding affinities and in vitro activity of different PADI4 variants. Data shows mean ± SEM of two independent replicates**

| Variant | $K_{50}$ DMSO (mM) | $K_{50}$ PADI4_11 (mM) | DMSO/ PADI4_11 | $K_D$_PADI4_11 (μM) | $K_D$_PADI4_3 (nM) |
|---|---|---|---|---|---|
| PADI4_WT | 0.20 ± 0.01 | 0.053 ± 0.002 | 4 | 0.457 ± 0.109 | 2.7 ± 0.5 |
| PADI4_D165A | 2.70 ± 0.46 | 1.82 ± 0.31 | 1.5 | 3 ± 2 | 61 ± 9 |
| PADI4_C166F | 0.34 ± 0.25 | 0.44 ± 0.27 | 0.9 | >10 | 3 ± 2 |
| PADI4_E167A | 0.43 ± 0.02 | 0.16 ± 0.02 | 2.7 | 0.7 ± 0.7 | 7 ± 0.7 |
| PADI4_D168A | 1.46 ± 0.42 | 0.50 ± 0.15 | 2.9 | 4 ± 1 | 20 ± 7 |
| PADI4_N373A | n/a | n/a | n/a | >10 | 83 ± 81 |
| PADI4_R394A | 0.62 ± 0.03 | 0.56 ± 0.08 | 1.1 | >10 | 8 ± 0.06 |

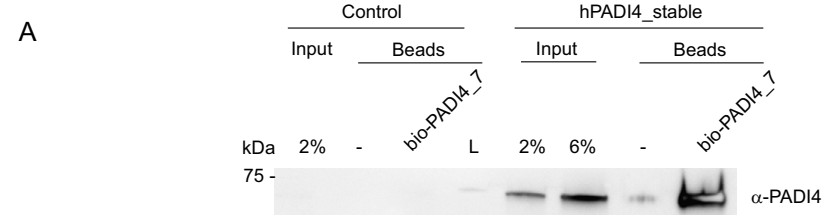

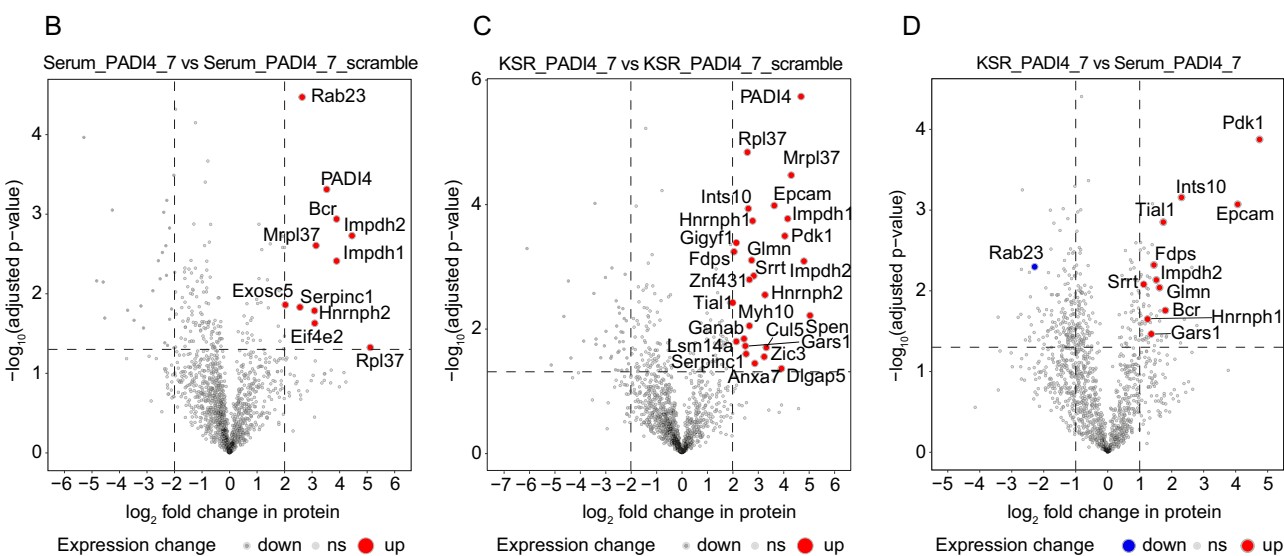

**Fig. 6 | Pull-downs with bio-PADI4_7. A** Immunoblot analysis for hPADI4. Pull-down assays were performed from hPADI4-stable or Control mES cells using bio-PADI4_7 as bait, or biotin-coated beads as negative control. Data show a representative experiment after two repeats. **B** Volcano plot representing the proteins enriched in bio-PADI4_7 versus bio-PADI4_7scr pull-downs from hPADI4-stable mES cells cultured in Serum conditions. Proteins highlighted in red were identified in all three replicates, with two sample Student's *t*-tests S0 = 0, both sides and *p* value ≤ 0.05 and Log2FC ≥2 for enrichment with the bio-PADI4_7 against bio-PADI4_7scr. **C** Volcano plot representing the proteins enriched in bio-PADI4_7 versus bio-PADI4_7scr pull-downs from hPADI4-stable mES cells cultured in KSR medium for 3 h. Proteins highlighted in red were identified in all three replicates, with two sample Student's *t*-tests S0 = 0, both sides and *p* value ≤ 0.05 and Log2FC ≥2 for enrichment with the bio-PADI4_7 against bio-PADI4_7scr. **D** Volcano plot representing the proteins enriched in bio-PADI4_7 pull-downs from hPADI4-stable mES cells stimulated with KSR- versus serum-containing medium for 3 h. Two sample Student's *t*-tests (S0 = 0, both sides and *p* value, FDR at 0.05; Log2FC ≥1) were used on proteins found in all three replicates in each condition. Proteins highlighted in red (for enrichment in KSR) or blue (for enrichment in Serum) are also enriched in bio-PADI4_7 over PADI4_7scr pulldowns with Log2FC ≥2.

tools based on PADI4_7 may be valuable tools for the molecular characterisation of PADI4 regulation. Proteins identified by Mass Spectrometry here point to potential co-factors or regulators of PADI4. For example, PADI4 activators or inhibitors may be enriched in KSR or Serum culture conditions, respectively, although regulatory proteins that exhibit equal binding under the two conditions may themselves be regulated through enzymatic activation and subsequently modulate PADI4 function. The data presented in Supplementary Data 2 may serve as a primer for understanding cellular PADI4 regulation.

The strong selectivity of our cyclic peptides for PADI4 over other PADI isozymes makes them ideal for disentangling the individual effects of these related proteins. One caveat of this selectivity is that they are also highly selective for hPADI4 over mPADI4, precluding their use in functional studies in mouse systems[71]. Focussed saturation mutagenesis libraries have previously been used to enhance the cross-species binding of RaPID-derived peptides[72]. This strategy would likely work well with PADI4_3, where small differences in multiple side chain-to-side chain interactions appear to be responsible for the selectivity (Fig. S3C). Interestingly, however, comparing the PADI4_11 binding site in human and mouse PADI4 reveals that F166 in mPADI4 (C166 in hPADI4) points directly into the binding site and would clash substantially, not only with side chains, but also with the helical backbone of the peptide, suggesting that an alternative scaffold will be required for developing a mouse-targeted PADI4 activator (Fig. S13E).

Whilst cyclic peptides are an attractive modality to develop potent and selective binders of challenging drug targets, progression as clinical candidates can be limited by their pharmacological properties, which often include poor metabolic stability, bioavailability and cell permeability[73]. In particular, their application to intracellular proteins has been limited. In this regard, PADI4_3 and PADI4_11 show surprisingly high cell permeability, of a level suitable for cellular experiments, demonstrating the potential of cyclic peptides to act as tools for intracellular protein targets. Additional enhancement of their stability and permeability will allow progress further in a translational context[74]. In the case of PADI4 in particular, a non-cell-permeable PADI4 inhibitor may be advantageous in the context of deregulation associated with cancer and inflammatory disease. In these contexts, it is proposed that citrullination of extracellular proteins by extracellular PADI4 promotes metastasis, or leads to increased inflammation and generation of autoantigens[23,75]. Indeed, given that the relationship between the physiological and pathological modes of PADI4 activation is not well understood, a non-cell-permeable PADI4 inhibitor might be valuable in targeting disease-associated PADI4 activity without disrupting its physiological intracellular roles. Towards this, simple modifications could likely be made to PADI4_3 to prevent its cellular uptake.

The work presented here describes the development and characterisation of a comprehensive toolbox of cyclic peptide modulators

of human PADI4 and methods for understanding its regulation within cells. The tools and methods developed here can be applied towards the study of the cell biological functions and biochemical mechanisms regulated by PADI4 in different physiological contexts and will, therefore, be applicable across different areas of cell biology and biomedical science. Importantly, they provide a platform for the development of cell modulatory and therapeutic agents applicable to the broad-ranging contexts of PADI4 deregulation, which include autoimmunity, cancer and age-related pathologies.

# Methods

## Ethical statement
Peripheral venous blood was isolated from consenting healthy adult volunteers according to approved protocols of the ethics board of the Francis Crick Institute and the Human Tissue Act. Written informed consent was obtained from participants. The study was performed in accordance with the ethical standards laid down in the 1964 Declaration of Helsinki and its later amendments.

## General information
Unless stated otherwise, reagents were purchased from commercial sources, including Merck, Nakalai, New England Biolabs (NEB) and Invitrogen. DNA oligonucleotides were purchased from Integrated DNA Technologies and Merck. Sanger sequencing was performed by Genewiz and whole plasmid sequencing by Plasmidsaurus or Full Circle.

## Protein constructs
The coding sequences of human *PADI2* (residues 1–663) and *PADI4* (residues 1–663) were amplified and cloned into the p28BIOH-LIC vector by ligation-independent cloning with the addition of a His6 tag between the vector encoded Avi-Tag and the Padi genes (Full encoded N-terminal tag sequence: MSGLNDIFEAQKIEWHEGSAGGSGHHHHHHGSGG). The coding sequences of *PADI1* (residues 1–663) and *PADI3* (residues 1–664) were amplified and cloned into pNIC28-BSA4 vector containing a His tag and a TEV protease cleavage site on the N-terminal site of the gene by ligation-independent cloning (Full encoded N-terminal tag sequence: MHHHHHHSSGVDLGTENLYFQSM). The coding sequence of mouse *Padi4* (residues 1–666) was amplified and cloned into the pET28a vector using NdeII (NEB) and XhoI (NEB) restriction enzymes. Mutagenesis of the plasmids was performed with the indicated primers (Table S4) using PfuTurbo Polymerase (Agilent) followed by DpnI (NEB) incubation at 37 °C for 1 h. All protein sequences were confirmed by Sanger sequencing.

## Protein production
To produce N-terminally biotinylated PADI4 (bio-His-PADI4), the plasmid was transformed into BL21-DE3 cells co-expressing biotin ligase (BirA) (TeBu Bio). Terrific broth was inoculated 1:100 with an overnight starter culture. At $OD_{600} = 0.6$, cells were induced with 0.2 mM IPTG and 20 μM D-biotin (Fluorochem Limited), and the temperature dropped to 18 °C. Cells were harvested after 16 h by centrifugation. Pellets were resuspended in binding buffer (50 mM HEPES, pH 7.5, 500 mM NaCl, 5 mM imidazole, 5% glycerol supplemented with DNase1 and 1xEDTA-free protease inhibitors (Roche)). Cells were lysed by sonication, and the clarified lysate was purified on a Ni-NTA 5 mL column using an ÄKTA Pure (Cytiva). Following washing with binding buffer, including a further 15 mM imidazole, the protein was step-eluted with a binding buffer containing 200 mM imidazole. Fractions containing bio-His-PADI4 were concentrated and further purified using an S200 size exclusion column (HiLoad 16/600 Superdex 200 (Cytiva)) in 50 mM HEPES, pH 7.5, 150 mM NaCl, 2 mM DTT, 5% glycerol. Pure fractions of bio-His-PADI4 were aliquoted, flash frozen and stored at −80 °C until use. Biotinylation was confirmed through streptavidin bead binding (Figure S1A).

His-tagged recombinant PADI enzymes were produced and purified as above, but from BL21(DE3) cells not co-expressing the biotin ligase, inducing only with IPTG.

## RaPID selections against PADI4
In vitro selections were carried out with bio-His-PADI4 following previously described protocols. Briefly, initial DNA libraries (including 6–12 degenerate NNK codons in a ratio 0.001 $NNK_{n=6}$:0.032 $NNK_{n=7}$:1 $NNK_{n=8}$:32 $NNK_{n=9}$:33 $NNK_{n=10}$:33 $NNK_{n=11}$:11 $NNK_{n=12}$) (see below for DNA sequence) were transcribed to mRNA using T7 RNA polymerase (37 °C, 16 h) and ligated to Pu_linker (Table S4) using T4 RNA ligase (30 min, 25 °C). First-round translations were performed on a 150 μL scale, with subsequent rounds performed on a 5 μL scale. Translations were carried out (30 min, 37 °C then 12 min, 25 °C) using a custom methionine(-) Flexible In vitro Translation system containing additional ClAc-L-Tyr-tRNA$^{fMet}_{CAU}$ (25 μM) or ClAc-D-Tyr-tRNA$^{fMet}_{CAU}$ (25 μM). Ribosomes were then dissociated by the addition of EDTA (18 mM final concentration, pH 8) and library mRNA reverse transcribed using MMLV RTase, Rnase H Minus (Promega). Reaction mixtures were buffer exchanged into selection buffer using 1 mL homemade columns containing pre-equilibrated Sephadex resin (Cytiva). Blocking buffer was added (1 mg/mL sheared salmon sperm DNA (Invitrogen), 0.1% acetyl-BSA final (Invitrogen)). Libraries were incubated with negative selection beads (3 × 30 min, 4 °C). Libraries were then incubated with bead-immobilised bio-His-PADI4 (200 nM, 4 °C, 30 min) before washing (3 × 1 bead volume selection buffer, 4 °C) and elution of retained mRNA/DNA/peptide hybrids in PCR buffer (95 °C, 5 min). Library recovery was assessed by quantitative real-time PCR relative to a library standard, negative selection and the input DNA library. The recovered library DNA was used as the input library for the subsequent round. Following completion of the selections, double indexed libraries (Nextera XT indices) were prepared and sequenced on a MiSeq platform (Illumina) using a v3 chip as a single 151 cycle reads. Sequences were ranked by total read numbers and converted into their corresponding peptide sequences for subsequent analysis (Supplementary Data 1).

Library DNA:

5′-TAATACGACTCACTATAGGGTTAACTTTAAGAAGGAGATATAC ATATG

(NNK)nTGCGGCAGCGGCAGCGGCAGCTAGGACGGGGGGCGGA AA

Three selections were carried out with differing buffer compositions:

Selection 1 and 3: 50 mM HEPES, pH 7.5, 150 mM NaCl, 2 mM DTT, 10 mM CaCl₂

Selection 2: 50 mM HEPES, pH 7.5, 150 mM NaCl, 2 mM DTT

Bead preparation:

For PADI4 immobilisation, bio-His-PADI4 was incubated with magnetic streptavidin beads (Invitrogen) (4 °C, 15 min to an immobilisation level of 0.9 pmol/μL beads) immediately before use in the selection. Biotin was added to cap unreacted streptavidin sites (25 μM final, 4 °C, 15 min). Beads were washed 3 × 1 bead volume selection buffer and left on ice for use in the selection. For selection 3, PADI4 immobilisation was carried out in the presence of Cl-amidine (200 μM, Cayman Chemicals) to covalently block the active site. Negative beads were prepared, similarly except that only selection buffer or selection buffer plus biotin (25 μM) were added to beads and following washing, these two variants were mixed.

## Mutational scanning of PADI4_3
A mutational scanning library was produced based on the PADI4_3 parent sequence in which each internal position was substituted for an NNK codon (Primers PAD4_3_F_NNK1-10, Table S4). An HA-tag was added c-terminal to the peptide to allow the purification of translated peptides before affinity panning with PADI4. A single round of

selection was performed with this library using a modified version of the protocol described for main selection 2 on a 5 μL scale in triplicate. Each 5 μL library replicate was derived from a single input library prepared on a 20 μL scale. Following the reverse transcription step, HA blocking solution was added (50 mM HEPES, pH 7.5, 0.2% acetylated BSA, 0.1% tween) and the sample incubated with 40 μL anti-HA magnetic beads (1 h, 4 °C), before washing (3 × 100 μL, 50 mM HEPES, pH 7.5, 0.05% tween) and elution with HA peptide (2 × 80 μL in 2 mg/mL in 50 mM HEPES, 300 mM NaCl, 10 mM CaCl₂, 2 mM DTT, 0.05% tween, 0.1% acetylated BSA, 1 mg/mL sheared salmon sperm DNA). For each replicate, 20 μL of eluted library was diluted to 200 μL in binding buffer (50 mM HEPES, 300 mM NaCl, 10 mM CaCl₂, 2 mM DTT, 0.05% tween, 0.1% acetylated BSA, 1 mg/mL sheared salmon sperm DNA) and added to bead-immobilised PADI4 (100 nM). Library and protein were incubated and washed under equilibrating conditions (3 × 200 μL, 10–12 h, RT). Libraries were eluted and recovered as described for the standard selection above and both input libraries after the HA purification and eluted libraries from PADI4 beads sequenced. Sequenced libraries were analysed using a custom Python script described previously[49]. An enrichment score was calculated for each peptide, where E = 0 means enrichment is the same as for the parent peptide sequence.

$$F_i = \frac{\text{reads}_i}{\sum \text{reads}} \quad e_i = \frac{F_{i,\text{output}}}{F_{i,\text{input}}} \quad E_i = \frac{e_i}{e_{\text{parent}}} \tag{1}$$

Where $i$ refers to a single peptide sequence (parent or variant). Errors were calculated as the standard deviation of $\log_2 E$ values for each replicate binding experiment.

## Peptide synthesis

Peptides were synthesised using NovaPEG Rink Amide resin as C-terminal amides by standard Fmoc-based solid-phase synthesis as previously described[76], using a Liberty Blue Peptide Synthesis System (CEM), a SYRO I (Biotage), a ResPep SLi multipep (CEM) or a Activotec P-11 peptide synthesiser. Following synthesis, the N-terminal amine was chloroacetylated by reaction with 0.5 M chloromethylcarbonyloxysuccinimide (ClAc-NHS) in dimethylformamide (DMF) (1 h, RT). The resin was washed (5 x DMF, 5 x dichloromethane (DCM)) and dried in vacuo.

Peptides were cleaved from the resin and globally deprotected with trifluoroacetic acid (TFA)/triisopropylsilane/1,2-ethanedithiol/H₂O (92.5:2.5:2.5:2.5) for 3 h at room temperature. Following filtration, the supernatant was concentrated by centrifugal evaporation and precipitated with cold diethyl ether. Crude peptides were resuspended in DMSO/H₂O (95:5) and, following basification with triethylamine to pH 10, were incubated with rotation for 1 h at room temperature. Peptides were then acidified with TFA and purified by HPLC (Shimadzu) using a Merck Chromolith column (200 × 25 mm) with a 10–50% gradient of H₂O/acetonitrile containing 0.1% TFA. Pure peptides were lyophilised and dissolved in DMSO for further use. Peptide stock concentrations were determined by absorbance at 280 nm based on their predicted extinction coefficients.

For biotinylation, an orthogonally protected Fmoc-Lys(mmt) was used. Following the addition of chloroacetyl-NHS to the N-terminal amino group, Lys(mmt) was selectively deprotected using 1% TFA and 5% triisopropylsilane in DCM (30 min, RT). The resin was then washed (5 x DCM, 5 x DMF) before being incubated with 20% diisopropylethylamine in DMF (10 min) and further washing (3 x DMF). The deprotected peptide was then allowed to react with 0.2 M NHS-biotin in DMF for 3 h. Global deprotection, cyclisation and purification were then carried out as described above.

For CAPA assays, chloroalkylation was performed using an orthogonally protected Fmoc-Lys(ivDde)-OH and an orthogonally protected Fmoc-Cys(mmt)-OH. Following the addition of chloroacetyl-NHS to the N-terminal amino group, Cys(mmt) was selectively

deprotected using 1% TFA, 5% triisopropylsilane in DCM (30 min, RT). Peptides were then cyclised on resin using 2% diisopropylethylamine in DMF overnight. Following cyclisation, Lys(ivDde) was deprotected using 2% hydrazine in DMF. The resin was then washed (3 x DCM, 3 x DMF) before on-resin coupling of chloroalkane carboxylic acid was performed on the lysine side chain using Hexafluorophosphate Azabenzotriazole Tetramethyl Uronium / diisopropylethylamine in DMF for 2 h. Global deprotection and purification were then carried out as described above. Chloroalkane carboxylate was synthesised as previously described[51].

Masses and LC–MS analysis of the different peptides are shown in Table S5 and Figs. S19–S65.

## Surface plasmon resonance

Single cycle kinetics analysis by SPR was carried out using a Biacore T200 or S200 and a Biotin CAPture kit, series S (Cytiva). Bio-His-PADI4 was immobilised on the chip to yield a response of ~2000 RU. 50 mM HEPES (pH 7.5), 150 mM NaCl, 2 mM DTT, 0.05 % Tween-20, 0.1% DMSO and variable CaCl₂ (as specified) was used as the running buffer and experiments performed at 25 °C.

Samples were run with 120 s contact time, and variable dissociation time (often 360 s). Data were analysed using the Biacore T200 analysis software; data was fitted to a 1:1 binding model accounting for baseline drift where necessary. Data are presented as the average ± standard deviation of at least two independent replicates.

## Isothermal titration calorimetry

Isothermal titration calorimetry (ITC) measurements were performed on a MicroCal PEAQ-ITC calorimeter (Malvern Panalytical). Peptides were diluted in buffer containing 50 mM HEPES, 150 mM NaCl, 0.1% DMSO and concentrations were calculated by A280 absorbance. Titrations were performed at 20 °C with peptide PADI4_11 (30–50 μM) in the cell and CaCl₂ (300–500 μM) in the syringe. Data were analysed using the MicroCal PEAQ-ITC analysis software supplied by the manufacturer using nonlinear regression with the 'One set of sites' model. For each experiment, the heat associated with ligand dilution was measured and subtracted from the raw data.

## COLDER assay

PADI4 activity was assessed using the COLDER assay[43] that measures the formation of urea-containing compounds (e.g. citrulline). The reaction was carried out in a 96-well plate with a final volume of 50 μl. His-PADI4 (50 nM) was incubated with different concentrations of peptides (100 μM to 0.3 nM) in the presence of CaCl₂ (from 10 mM to 0.001 mM as specified) for 10 min. The reaction was started by the addition of 10 mM $N^\alpha$-Benzoyl-L-arginine ethyl ester hydrochloride (BAEE, Merck). After 30 min, the reaction was quenched with EDTA (50 mM final concentration). For colour development, 200 μl of COLDER solution, consisting of 1 volume of solution A (80 mM diacetyl monoxime/2,3-butanedione monoxime (Merck) and 2 mM thiosemicarbazide (Acros Organics)) and 3 volumes of solution B (3 M H₃PO₄, 6 M H₂SO₄ and 2 mM NH₄Fe(SO₄)₂) were added to each well and the mixture was heated at 95 °C for 15 min. After cooling to RT, absorbance at 540 nm was measured, and citrulline concentration was determined using a standard curve of citrulline standards. All studies were performed within the linear range of PADI4 activity, and data analysis was performed using Prism GraphPad (Dotmatics). Data are presented as the average ± standard error of the mean from at least two independent replicates.

Activity assays with other PADI enzymes were performed following the same protocol using the following enzyme concentrations: hPADI1, 100 nM, hPADI2, 50 nM, hPADI3, 250 nM and mPADI4, 50 nM.

## Cryo-electron microscopy

CryoEM samples were prepared of PADI4 at 18 μM and peptide at 90 μM, supplemented with 0.1% (w/v) ß-octylglucoside to reduce

preferential orientation. Samples were applied to 200 mesh Quantifoil R2/2 grids, before blotting and plunge freezing in liquid ethane using a Vitrobot Mk III. Data were collected using EPU software on a Thermo Scientific Titan Krios microscope operating at 300 kV, equipped with a Falcon 4i and Selectris energy filter operating in zero loss mode with 10 eV slit width. Movies of 6.1 s were acquired in .eer format with a pixel size of 0.95 Å and a total accumulated dose of 28.0 e/Å$^2$. The motion in the movies was corrected using MotionCor2 implemented in RELION[77], subdividing the .eer file in 36 fractions, each with 0.78 e/Å$^2$. Contrast Transfer Function was estimated using CTFfind4. Particles were picked using Topaz and 2D classified using cryoSPARC v4[78]. Particles belonging to classes with well-defined secondary structure were selected, and an initial 3D model was calculated using the ab initio reconstruction in cryoSPARC. These selected particles were refined in RELION 3D Autorefine prior to Bayesian polishing. Polished particles were classified using 3D classification in RELION. Particles belonging to the best-defined class were refined using non-uniform refinement in cryoSPARC coupled to CTF refinement. The maps obtained had their local resolution estimated using blocres implemented in cryoSPARC, followed by filtering by local resolution and global sharpening in cryoSPARC. A summary of the cryoEM data processing is shown in Supplementary Fig. 17.

Model building was carried out using the crystal structure PDB: 1WD9 as a starting reference. The model was adjusted and peptides built using Coot[79] coupled with rounds of real-space refinement and validation in PHENIX[80]. CryoEM figures were generated using UCSF ChimeraX[81] (version 1.6.1.).

### Chloroalkane penetration assay (CAPA)

CAPA assays were performed following the protocol published in ref. [51]. In brief, 40,000 HaLo- GFP-mito HeLa cells were seeded in a 96-well plate. Twenty-four hours later, cells were treated with different concentrations of chloroalkane-tagged peptides (100 µM to 200 nM for PADI4_**3** and PADI4_**3i** and 50 µM to 100 nM for PADI4_**11** and PADI4_**11i**) for 4 h in Opti-MEM. Media containing peptides was then removed, and cells were incubated with Opti-MEM (Gibco) for 15 min. After that time, the wash was removed, and cells were chased with HaLo-Tag TMR ligand (5 µM, Promega) in Opti-MEM for 15 min. The ligand was then removed, and cells were incubated with Opti-MEM for 30 min. Cells were then trypsinised, resuspended in phosphate-buffered saline (PBS) with 0.5% FBS and analysed using a benchtop flow cytometer (MACSQuant® VYB Flow cytometer Milteny Biotec). For each well, 10000 GFP positive single cells were analysed (gating workflow shown in Supplementary Fig. 18), and TMR mean fluorescence was normalised against a positive control (cells treated with HaLo-Tag TMR ligand but no peptide) and a negative control (non-treated cells). Analysis was performed using FlowJo (LCC) software and Prism GraphPad (Dotmatics). Data were presented as the average ± standard error of three independent replicates.

### Cell culture and cell line generation

HaLo-GFP-mito HeLa cells were grown in DMEM (Gibco) supplemented with 10% foetal bovine serum (FBS), penicillin/streptomycin (100 µg/ml) and puromycin (1 µg/ml) at 37 °C, 5% CO$_2$. HaLo-GFP-mito HeLa cell line[82] was profiled by short tandem repeat analysis (STR) and tested negative for mycoplasma by the Cell Services Technology Platform at the Francis Crick Institute. E14 –mouse embryonic stem cells (Cambridge Stem Cell Institute, UK) were cultured in plates coated with 1% gelatine in GMEM supplemented with 10% foetal calf serum (FCS) batch-tested for ES cell culture (Gibco), 0.1 mM non-essential amino acids, 2 mM L-glutamine, 1 mM sodium pyruvate, 0.1 mM β-mercaptoethanol and 50 ng/mL leukaemia inhibitory factor (LIF) (ESGRO, Millipore). For PADI4 activation, cells were cultured in GMEM containing 10% knockout serum replacement (KSR, Life Technologies), 1% FCS batch-tested for ES cell culture (Gibco), 0.1 mM non-essential

amino acids, L-glutamine, 1 mM sodium pyruvate, 0.1 mM β-mercaptoethanol, 50 ng/mL LIF (ESGRO, Millipore), 1 µM PD0325901 (AxonMedChem) and 3 µM CHIR99021 (AxonMedChem). For the generation of E14 mES stably expressing human PADI4 (hPADI4-stable), human PADI4 cDNA was inserted into ES E14 cells using the piggyBac transposon system[83]. The Gateway system was used to clone human PADI4 into the piggyBac vector using primers PADI4_AttB1_F and PADI4_AttB2_R (Table S4). pPB-CAG-CTRL (empty vector) or pPB-CAG-hPADI4 vectors (1 mg) were transfected with the piggyBac transposase (pPBase) expression vector, pCAGPBase (2 mg), by lipofection according to the manufacturer's instructions (Lipofectamine 2000, Invitrogen). E14 cells constitutively expressing the hygromycin resistance gene and human PADI4 were selected and expanded in media containing 200 mg/ml hygromycin.

### High content microscopy

E14 cells were seeded onto 96-well PerkinElmer cell carrier ultra plates (Phenoplates) coated with 0.1% gelatine, at 25,000 cells per well and grown overnight at 37 °C, 5% CO$_2$. Outer wells were excluded but medium was added to minimise any edge effects. Serum and KSR/2i media were freshly made on the day of the treatment. After treatment with KSR/2i and cyclic peptides, cells were washed with PBS, fixed with ice-cold 100% methanol for 5 min at RT and washed twice with PBS. Cell permeabilisation was performed with 0.1% Triton X-100 in PBS for 10 min at RT, and blocking was done in 5% BSA, 0.1% Triton X-100 in PBS at 37 °C overnight, covering the plate with parafilm to minimise evaporation. Cells were stained with anti-H3Cit2 antibody (Abcam ab176843; 1:150 in blocking buffer) on a rocking platform for 1 h at RT, washed (3 x PBS) and stained with secondary antibody (Alexa488-conjugated cross-adsorbed goat anti-rabbit, 1:200 in blocking buffer) on a rocking platform for 2 h at RT. Cells were washed (PBS, 3 × 5 min) on a rocking platform and stained with 4',6-diamidino-2-phenylindole dihydrochloride (DAPI) (Life Technologies) staining (2 µM) for 10 min at RT. After two final washes (PBS, quick changes), the cells were imaged. Technical triplicates were included for each condition on each plate. A minimum of nine images were captured per well on either an InCell6000 (GE/Cytiva) or an ImageXpress Confocal HT.ai (Molecular Devices) high-content imaging system through a Nikon PlanApochromat 10x/0.45NA objective lens. On both systems, laser-based auto-focusing technology was applied to ensure all images are in focus. The DAPI channel was captured by illuminating samples with a 405 nm laser line, and acquiring the signal through a 452/45 nm emission filter. The A488 image was acquired by using the 475 nm laser line on the HT.ai, or the 488 nm laser line on the InCell6000 and capturing the emission through a 525/28 nm emission filter. All images were acquired at 2048 × 2048 pixels (1 × 1 binning) in wide-field mode. Image analysis was performed using a CellProfiler[84] pipeline. The IdentifyPrimaryObject module was used to detect nuclei in the DAPI image and create a binary mask of individual, segmented nuclei. A MeasureObjectSizeShape module was then incorporated to assess nuclear morphology, and the MeasureObjectIntensity module was applied to measure the mean A488 intensity within the nuclei masks as a readout of nuclear H3Cit levels. The data were exported using the ExportToSpreadsheet module. Data were normalised to the basal Serum-only condition. Technical triplicates and data from biological replicates were merged and plotted using GraphPad Prism. The average of each dose was taken and plotted for the EC$_{50}$ calculations. Outliers were removed using the GraphPad prism ROUT programme.

### Calcium imaging

Calbryte™ 520 AM dye (https://www.aatbio.com/products/calbryte-520-am) was used for quantification of calcium influx into cells. Cells were seeded onto 10 mm optical imaging plates at 10$^5$ cells per dish and allowed to attach overnight. Cells were stained with 10 µM Calbryte dye in normal growth medium for 1 h, 37 °C, 5% CO$_2$, washed

once with cell culture medium and imaged using a Nikon Wide-field microscope. A time series was performed, with a 488 laser image taken every second. After 1 min of imaging (baseline readings), PADI4_11 (25 μM), PADI4_11i (25 μM) or calcium ionophore (10 μM, as positive control) were added, and cells were imaged for a further 60 min. Analysis was done using imageJ —region of interest (ROI) were taken for each cell at each time point using ROI manager. Background values (ROI with no cells) were taken at each time point and removed from the same background reading (i.e. ROI at 1 second with cells minus ROI at 1 s background value). An average of at least ten cells was taken for each condition and each average data point was divided by the initial average data point for that cell, to obtain overall response.

## Cytotoxicity assays

Cells were treated with increasing concentrations of PADI4_11 and PADI4_11i (15 μM, 25 μM or 40 μM) for 1 h at 37 °C, 5% $CO_2$. Cells were detached, spun at $443 \times g$ for 3 min and washed once in PBS, followed by a further spin at $443 \times g$ for 3 min. The cells were then resuspended in PBS + 0.1% BSA. Samples were kept on ice. Propidium iodide (https://www.thermofisher.com/order/catalog/product/V35118) was added at 1.25 μg/mL, and cells were incubated for 5 min on ice. Cells were then analysed using a Fortessa flow cytometer. Live cell imaging was performed using an Incucyte® imaging system (Sartorius), using Incucyte® Cytotox Green Dye as per the manufacturer's instructions.

## Human neutrophils from peripheral blood healthy donors

Peripheral venous blood was isolated from consenting healthy adult volunteers in heparin tubes as previously described[85]. Briefly, blood samples were on Histopaque 1119 (Sigma-Aldrich) and then centrifuged for 20 min at $800 \times g$. The plasma was collected and centrifuged for a second time for 20 min at $800 \times g$. The neutrophil layer was collected and washed in Hyclone Hank's Balanced Salt Solution (HBSS) -Ca, -Mg, -Phenol red (GE Healthcare) supplemented with 10 mM HEPES (Invitrogen) 0.1% FBS and further purified with a discontinuous Percoll (GE Healthcare) gradient consisting of layers with densities of 85, 80, 75, 70 and 65% and centrifuged for 20 min at $800 \times g$. Neutrophil-enriched layers were then collected and washed, and neutrophil purity was assessed using flow cytometry.

## Cholesterol crystal stimulation and NET preparation

Cholesterol (Sigma-Aldrich) was solubilised in 95% ethanol to a final concentration of 12 mg/mL at 65 °C whilst shaking. Cholesterol crystals were then formed by placing tubes on ice for 30 min, followed by a further five freeze-thaw cycles. Crystals were then spun down at 4 °C for 15 min at $16,200 \times g$ and resuspended in PBS to a final concentration of 5 mg/ml.

About $1 \times 10^6$ human blood neutrophils were isolated as described in the "Human neutrophils from peripheral blood healthy donors" section. Neutrophils were seeded in 6 well plates in HBSS +Ca, +Mg, -Phenol red (GE Healthcare) containing 10 mM HEPES (Invitrogen), left to settle for 30 min, then pre-incubated with either 50 μM PADI4_3, PADI4_3i or DMSO (0.5%) for 1 h. Cholesterol crystals were then added to wells to a final concentration of 1 mg/mL, and plates were spun down at $443 \times g$, RT for 3 min, to decant the crystals. Supernatants were carefully removed the following morning, and 250 μL restriction enzyme mix containing 10 U/mL of BseRI, PacI, NdeI, AflII (NEB), 10 μM neutrophil elastase inhibitor (GW311616A) and 10 μM cathepsin G inhibitor I (Sigma-Aldrich) in CutSmart buffer (NEB) was added to wells for 30 min at 37 °C to partially digest NETs. NET DNA concentration was determined by Quant-iT Picogreen dsDNA reagent as per the manufacturer's instructions (Thermo Fisher Scientific).

## Time-lapse imaging and NET area quantification assay

About $5 \times 10^4$ human neutrophils were seeded in a black 96-well plate (PerkinElmer) in HyClone HBSS +Ca, +Mg, - Phenol red (GE Healthcare)

containing 10 mM HEPES (Invitrogen). Cells were pre-incubated with the corresponding peptide for 60 min prior to the start of the experiment. DNA of live cells was stained with 4 μg/mL Hoechst (membrane permeable; Thermo Scientific) and DNA from dead cells with 0.2 μM Sytox-green (membrane impermeable; Invitrogen). Cells were stimulated with 100 nM PMA, 5 μM Ca-Ionophore or 0.1 mg/mL cholesterol crystals at the start of the microscope acquisition. Plates stimulated with cholesterol crystals were centrifuged at $300 \times g$ for 5 min at RT immediately prior to microscope acquisition. The cells were imaged on an inverted Nikon wide-field microscope system at 37 °C and $CO_2$ (5%). Four fields of view were acquired per well every 15 mins for 10–15 h using a 40x objective. NETs were differentiated from necrotic cells by the size of the sytox signal using FIJI[86]. Analysis of DNA area was assessed by Sytox-green signal after 8 h cholesterol crystal stimulation using FIJI as described in ref. 85. Area measurements were distributed into bins of increasing area sizes and plotted as a percentage of the total number of Sytox-green positive (dead) cells.

## Immunostaining and confocal microscopy of NETosis steps

About $5 \times 10^4$ sorted neutrophils were seeded in 24-well cell culture plates containing glass coverslips in HyClone HBSS +Ca, +Mg, - Phenol red (GE Healthcare) supplemented with 10 mM HEPES (Invitrogen). Neutrophils were stimulated with 5 μM Ca-Ionophore (Sigma) and fixed for 20 min at room temperature with 2% paraformaldehyde at RT. Cells were then washed and permeabilized with 0.5% Triton X-100 in PBS. Non-specific binding was blocked with 2% BSA (Sigma) and 2% donkey serum (Sigma) in PBS. Samples were stained with DAPI (Life Technologies), the rabbit anti-CitH3 (Abcam; ab5103 1:500) and donkey anti-rabbit alexa488 (Invitrogen; A21206). Fluorescence imaging was performed on a Leica TCS SP8 inverted confocal microscope using sequential scan in between frames mode with a 20x objective.

## Immunoblot analysis

Neutrophil samples were boiled in a sodium dodecyl sulphate (SDS) buffer containing dithiothreitol (DTT) and run on a polyacrylamide gel electrophoresis (SDS-PAGE) using Criterion TGX precast gels (Any-KD; Bio-Rad Laboratories). Proteins were then transferred to a PVDF membrane (Bio-Rad Laboratories) via semi-dry transfer. The membrane was blocked with 5% BSA (Fisher Scientific) in Tris-buffered saline with 0.1% Tween-20 (TBS-T). Levels of citrullinated histone H3, total H3 and myeloperoxidase (MPO) were assessed by incubating membranes in either anti-histone H3 citrulline R2 + R8 + R17 (Abcam; ab5103 1:1000), anti-histone H3 (Millipore; 07-690 1:10000) or anti-MPO (DAKO; A0398 1:2500) overnight at 4 °C, respectively. For cell lysate-based assays and pull-down assays, membranes were incubated as above with anti-hPADI4 (Abcam; ab50332; 1:1000 in 5% BSA/TBS-T), anti-Bcr (Cell Signalling #3902S; 1:1000 in 3% milk/TBS-T) and anti-Impdh2 (Cell Signalling #57068S; 1:1000 in 3% milk/TBS-T). Membranes were washed in 0.1% TBS-T and subsequently incubated with goat anti-rabbit HRP or anti-goat (Fisher Scientific) for 1 h, followed by final washes in 0.1% TBS-T. Membranes were imaged with enhanced chemiluminescent substrate (ECL) (Thermo Fisher) on a Bio-Rad imager.

## Pull-downs with biotinylated cyclic peptides

Four sets of pull-downs were performed in triplicate: bio-PADI4_7 and bio-PADI4_7scr, in Serum or KSR, medium-treated PADI4-stable mES cells. Streptavidin M-280 Dynabeads (Thermo Fisher Scientific) were conjugated with bio-PADI4_7 or bio-PADI4_7scr by incubation for 45 min at room temperature with gentle rotation and washed three times in PBS containing 0.1% BSA, before they were used for pull-downs. One 10 cm dish of cells was used per experimental replicate. Cells were incubated in freshly made serum-containing medium (GMEM supplemented with 10% foetal calf serum (FCS) batch-tested for ES cell culture (Gibco), 0.1 mM non-essential amino acids, 2 mM L-

glutamine, 1 mM sodium pyruvate, 0.1 mM β-mercaptoethanol and 50 ng/mL leukaemia inhibitory factor (LIF) (ESGRO, Millipore)) or KSR-containing medium (GMEM containing 10% knockout serum replacement (KSR, Life Technologies), 1% FCS batch-tested for ES cell culture (Gibco), 0.1 mM non-essential amino acids, L-glutamine, 1 mM sodium pyruvate, 0.1 mM β-mercaptoethanol and 50 ng/mL LIF (ESGRO, Millipore)) for 3 h at 37 °C, 5% $CO_2$. Cells were washed in cold PBS and harvested by scraping directly in lysis buffer (10 mM Tris/HCl pH 7.5, 150 mM NaCl, 0.2% NP-40, with freshly added Benzonase (250 U), 2.5 mM $MgCl_2$ and protease inhibitors) and incubated at 4 °C with gentle rotation for 30 min. Lysates were centrifuged at 18800 × $g$ for 10 min in a cold microfuge, and insoluble chromatin pellets were discarded. The lysates were diluted with Dilution/Wash buffer (10 mM Tris/HCl pH 7.5, 150 mM NaCl, with freshly added protease inhibitors) and incubated with peptide-conjugated beads at 4 °C with gentle rotation overnight. The beads were separated on a magnetic rack and washed (3 x Dilution/Wash buffer). The bound proteins were eluted in NuPAGE LDS sample buffer with boiling at 95 °C for 5 min, then run ~10 mm into an SDS-PAGE gel. The gel was stained with Imperial Protein Stain (Thermo Fisher) as per the manufacturer's instructions.

## Mass spectrometry analysis

Pull-downs were performed as described above (three replicates per condition). Gel pieces were excised and destained, then proteins were reduced by dithiothreitol, alkylated by iodoacetamide and digested overnight with trypsin (sequence modified, Promega). An aliquot of the digest was analysed by LC–MS/MS on a Q Exactive™ Plus Orbitrap™ mass spectrometer equipped with an UltiMate 3000 RSLCNano LC system (both from Thermo Fisher). Peptides were loaded by 0.1% formic acid (FA)/$H_2O$ to a trap column (100 μm i.d. × 5 cm), and then separated on an analytical column (100 μm i.d. × 20 cm) at a linear gradient from 2 to 35% MeCN/0.1% FA in 60 min. Both trap and analytical columns were self-packed with C18 particles (3 μm Reprosil C18AQ) Dr. Maisch, Babraham Proteomics Facility). The MS analysis used a standard data-dependant acquisition method. The MS1 survey scan was between m/z 350–1800 at 70,000 resolution, maximum injection time at 50 ms and Automatic Gain Control (AGC) at $3 × 10^6$. The top 20 multiply charged ions ($z$ = 2–4) with above $1 × 10^4$ intensity threshold were isolated and fragmented with a normalised collision energy at 27. The isolation window was set at 1.2 Th. The settings for the MS/MS acquisition were: the resolution at 17,500, AGC at $1 × 10^5$ and maximum injection time at 200 ms. The dynamic exclusion window was 30 s.

The raw files were processed in Proteome Discoverer 2.5 (Thermo Fisher) using the Sequest HT search engine to search against mus musculus proteome database (UniProtKB, version June 2022), human PADI4 (Q9UM07) and common contaminate protein database. Search parameters were: trypsin (full specificity) with two maximum missed cleavage sites, mass tolerances at 10 ppm for the precursor, and 0.02 Da for the fragment ions; oxidation (M) and acetylation (Protein N-terminus) as dynamic modifications, and Carbamidomethyl (C) as static modification. Peptides were validated by Percolator with a $q$ value set at 0.01 for the decoy database search, and only high confident PSMs (Peptide Spectrum Matches) were considered. Protein FDR was set at 0.01. The label-free quantification settings used both unique and razor peptides. Only master proteins with high-confidence peptides were reported.

All contaminants, proteins with no abundance value in any samples, and proteins identified by only 1 peptide were removed before further analysis in Perseus (v 2.0.11.0) (https://maxquant.net/perseus/)[87]. Before statistical analysis, missing values were imputed using the default setting in Perseus. The protein abundances were normalised by the median for each sample, and log2 was transformed. For the comparison of bio-PADI4_7 against bio-PADI4_7scr in both culture conditions (KSR and Serum) and of bio-PADI4_7 in KSR against Serum conditions were used

for the two sample t-test, was used after missing values were imputed used the default settings in Perseus. The Student's t-tests were based on $p$ value (S0 = 0, both sides, threshold at 0.05). Further filtering for true hits was based on $p$ value ≤ 0.05 (i.e. −Log10 $p$ value ≥1.301) and Log2FC ≥2 for enrichment with the bio-PADI4_7 against bio-PADI4_7scr. For the bio-PADI4_7 in KSR against Serum conditions, further filtering for regulation was based on $p$ value ≤ 0.05, Log2FC ≥1, including only proteins that were identified as true hits in the former comparisons. The full dataset, with and without imputation, is provided in Supplementary Data 2.

### Reporting summary

Further information on research design is available in the Nature Portfolio Reporting Summary linked to this article.

## Data availability

Unless otherwise noted, all data supporting the results of this work are available in the article, supplementary information, and source data files, or accessible via the public repositories listed herein. CryoEM maps generated in this study have been deposited in the Electron Microscopy Data Bank (EMDB) under the accession codes EMD-19011 (PADI4:PADI4_3) and EMD-19012 (PADI4:PADI4_11). The atomic coordinates have been deposited in the Protein Data Bank (PDB) under the accession codes 8R8U (PADI4:PADI4_3) and 8R8V (PADI4:PADI4_11). Mass spectrometry data was deposited to PRIDE depository under the accession code PXD048807. NGS Sequencing data has been uploaded in the NIH Short Read Archive (SRA) under the accession number PRJNA1167468. Source data are provided with this paper.

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

## Acknowledgements

p28BIOH-LIC was a gift from Cheryl Arrowsmith (Addgene plasmid # 62352; http://n2t.net/addgene:62352; RRID:Addgene_62352). We thank Laura Masino and Simone Kunzelman from the Structural Biology STP for their help with the isothermal titration calorimetry and surface plasmon resonance experiments. We also thank the Crick Advanced Light Microscopy, the Flow Cytometry, the Proteomics and the Cell Service Scientific Technology Platforms, and Kranthi Yadav (Babraham Proteomics Facility) for technical assistance. This work was supported by the Francis Crick Institute, which receives its core funding from Cancer Research UK (CC2030 and CC2089), the UK Medical Research Council (CC2030 and CC2089) and the Wellcome Trust (CC2030 and CC2089), funding to L.J.W. from the European Union's Horizon 2020 research and innovation programme under Marie Skłodowska-Curie Grant Agreement 657292, by the Babraham Institute, which receives its core funding from the UK Biotechnology and Biological Sciences Research Council (BBS/E/B/000C0421) and by a Wellcome Trust/Royal Society Sir Henry Dale Fellowship to MAC (105642/Z/14/Z). I.V.A. was funded by a Sir Henry Wellcome Trust fellowship (SHWF 222825/Z/21/Z).

## Author contributions

M.T.B., R.W., T.C., I.V.A., T.S., M.A.C. and L.J.W. planned and executed experiments and analysed data; J.A., M.C., R.M.M., D.J. and S.F. conducted experiments; D.B. carried our CryoEM analyses; H.O. developed the analysis pipeline and carried out bioinformatic analyses of imaging data; L.Y. conducted Mass Spectrometry analyses; D.O. consulted on Mass Spectrometry; S.W. consulted on imaging work; T.C., M.A.C. and L.J.W. conceptualised the project; V.P., H.S., M.A.C. and L.J.W. obtained funding and supervised the work; M.T.B., M.A.C. and L.J.W. wrote the manuscript with help from all the authors.

## Funding

## Competing interests

M.T.B., R.W., T.C., M.A.C. and L.J.W. are inventors on UK Patent application No. 230979.0, which has been filed with respect to methods

that modulate cellular reprogramming through enhancing the activity of the PADI4 enzyme. The remaining authors declare no competing interests.
