## [Peer Review File · Nature Communications]

A cyclic peptide toolkit reveals mechanistic principles of peptidylarginine deiminase IV regulationREVIEWER COMMENTS

Reviewer #1 (Remarks to the Author):

Deregulation of the enzyme peptidylarginine deiminase IV (PADI4) is implicated in the development of a number of diseases, including autoimmunity, cancer, atherosclerosis and age-related tissue fibrosis. There are currently no clinically relevant inhibitors. In the present study, the mRNA-display-based RaPID system was used to screen a large library of cyclic peptides for high-affinity, conformation-selective binders, resulting in PADI4_3, an inhibitor specific for the active conformation of PADI4, PADI4_7, an inert binder, and PADI4_11, an activator. These peptides represent a valuable set of probes to study the structure and function of PADI4, which are already utilised in this study, for example to investigate the role of human PADI4 in NETosis. Regrettably, their high specificity for the human enzyme will preclude their use in mouse models.

Overall, this manuscript describes a substantial body of work and takes advantage of the relatively new RaPID approach. I have a number of comments that should be addressed before acceptance, as follows:

The Introduction is well written and makes a strong case for the development of PADI4 ligands, both inhibitors and activators. Nonetheless, I felt that that it was too long, and could be trimmed to half to 2/3 of its current length without loss of clarity.

Figure S7C shows the results of a CAPA assay indicating that PADI4_3 and PADI4_3i enter cells. PADI4_11 and PADI4_11i are also cell permeable, as documented in Figure S9E. The Discussion (p. 17) acknowledges that PADI4_3 and PADI4_11 show surprisingly high cell permeability, but this important point is not further elaborated. Given the challenges associated with getting peptides and macrocycles across cell membranes, the apparent permeability of these peptides should be further characterised. While this may be beyond the scope of the current manuscript, the desirability of further work should be mentioned. Is it known if the peptides are taken up initially into endosomes and then released into the cytoplasm?

The pull-down results in Figure 6 identify a number of binders of bio-PADI4_7. These are discussed briefly in the current manuscript, but clearly represent an interesting avenue for further study.

Minor comments:

p. 5: 'Sheared salmon sperm DNA was included in the selection buffers to reduce high levels of non-specific nucleic acid binding to PADI4 which were observed during initial optimisations.' Presumably this is based on literature precedent, in which case a reference should be provided.

p. 7: 'Most notably H640, which...' This is not a sentence.

p. 9: CAPA should be defined at first use.

p. 21: 'Peptides were synthesised using NovaPEG Rink Amide resin as C-terminal amides by standard Fmoc-based solid phase synthesis as previously described...' A reference is needed.

p. 21: 'Pure peptides were lyophilised and dissolved in DMSO for further use.' Is this because they were poorly soluble in water?

p. 23: 'For colour development, 200 µl of COLDER solution, consisting on 1 volume..' Replace on with of

p. 25: 'for lysated-based' lysate-

p. 25: 'A minimum of nine images were captured..' was

p. 29: bovine serum albumin should be abbreviated

Table 1: the heading should indicate to which target the affinities refer.

Figure 5D: the dark shading makes it difficult to identify the residues; I suggest using lighter shading colours.

Standard abbreviations for time should be used throughout.

Reviewer #2 (Remarks to the Author):

Bertran et al (2024) A cyclic peptide toolkit reveals mechanistic principles of PAD4 regulation

The manuscript by Bertran et al describes the use of the RAPID approach to identify a series of cyclic peptides that bind selectively to human PAD4. These cyclic peptides fall into one of three categories: i) inhibitors of PAD4 activity; ii) compounds that allosterically decrease the concentration of calcium that is required to activate PAD4; and iii) peptides that bind to PAD4 and can be used to isolate this protein from complex mixtures. All three types of compounds are potentially useful. For example, the authors disclose compound 3 which binds to the active site of PAD4 to inhibit its in vitro activity with an IC₅₀ of ~50 nM. Impressively, the authors obtained the cryoEM structure of PAD4 bound to this cyclic peptide and identified key interactions that are important for its potency and selectivity. The finding that compound 11 allosterically decreases the calcium dependence of PAD4 is also impressive because it provides strong data supporting the notion that an interacting protein can modulate the calcium dependence of the protein to decrease the concentration of calcium required to activate PAD4 to physiologically relevant levels. Although 11 does not decrease the concentration to <1 micromolar the reduction is nonetheless significant. The development of the biotinylated version of compound 7 is also important because it will facilitate the isolation of PAD4 from a variety of complex mixtures. Notably, the authors use this compound to identify potential interacting proteins. Future work will be needed to validate these potential interacting proteins. Overall, this manuscript represents a significant advancement and provides tools that will be used by the wider community. This reviewer recommends acceptance once the authors address the following.

1. The authors note several times that GSK484 binds the inactive form and that it would be better to have an inhibitor that binds the active form. I think this description of GSK484 is unfair and could be improved as this compound is an allosteric inhibitor that preferentially binds to the calcium free form of the enzyme, stabilizing it, such that calcium cannot bind and activate the enzyme. Since the calcium free and calcium bound forms of PAD4 are in rapid equilibrium, GSK484 will inhibit PAD4 in the presence of calcium. It is just that at very high levels of calcium less inhibition is observed.
2. Several GSK484 derivatives have been disclosed by BMS, Gilead, and Jubilant Therapeutics and this work should be discussed. See *Curr. Op. Chem. Biol.* 75, 102313 for references.
3. PAD1 and PAD2 specific inhibitors have been disclosed [PMC5477668 *Angew Chem Int Ed Engl.* 58, 12476]. These works should be referenced.
4. The PAD community, especially the rheumatology community, routinely refers to the proteins as PADs and the gene as PADI. I know they are annotated as PADI in uniprot but when the vast majority of papers refer to the enzymes as PADs, I think the authors should follow convention.
5. The authors should determine whether compound 11 is a PAD4 substrate.
6. The authors note that compounds 11 and 12 can activate PAD4 activity by 36% and 12% respectively. Is that increase in activity statistically significant?
7. Fig 2a. GSK488 should be GSK484
8. Fig 2e. define the color scale in the figure legend.
9. The authors should highlight the H4R enrichment data and address whether compound 3H4R would be a substrate.
10. The PADI4_3 nomenclature is a little clumsy. It would be simpler if the authors used bold numbers, e.g. 3, 7, 11, etc to refer to the compounds.
11. Panel 4F appears to quantify the data in 4E but the data does not seem to match up well. For example, with 11i there is a significant signal at 30 uM that is not apparent in 4F.
12. The authors used 50 nM of PAD4 in their inhibition assays and their most potent compound has an IC₅₀ of ~ 50 nM. Since they are titrating the enzyme, does the IC₅₀ for 3 underestimate the true IC₅₀? This seems likely based on their SPR data. The authors should comment on this issue in the manuscript.
13. Figure S14 panel G. This panel is unlabeled. Is this wtPAD4 or a specific mutant.
14. Regarding the pulldown data with bio7, it would be nice to determine whether any of the identified proteins are truly PAD4 interacting proteins by western blotting.

Reviewer #3 (Remarks to the Author):

The manuscript by Bertran is very well prepared and presented and will make an important contribution to the field. The results are well supported by the data and a broad range of methods have been used to identify and validate the new inhibitors. I am not an expert on the system or the activity assays but they appear to be robust. My experience is within structural biology and the data appear robust and the data processing looks appropriate.

I do not have any significant edits to suggest nor any additional experiments. The one suggestion I had which would aid in robustness is that the authors should show the data for the structures i.e. the map. The fitted model is shown but there are no figures which show the correlation of map and model. At the resolution there will be some ambiguity in the fit in parts of the model. Therefore it would be good to have some examples of the correlation between map and model. This does not need to be done for all figures as it can reduce the clarity. For example in Supp figure 13 what is the map quality like in panels B to E?

REVIEWER COMMENTS

We thank all the reviewers for their positive evaluation of our manuscript. We are grateful for their useful comments and valuable insights on our manuscript and have edited it accordingly. We provide a point-by-point response below.

Reviewer #1 (Remarks to the Author): changes are highlighted in the manuscript in **yellow**

Deregulation of the enzyme peptidylarginine deiminase IV (PADI4) is implicated in the development of a number of diseases, including autoimmunity, cancer, atherosclerosis and age-related tissue fibrosis. There are currently no clinically relevant inhibitors. In the present study, the mRNA-display-based RaPID system was used to screen a large library of cyclic peptides for high-affinity, conformation-selective binders, resulting in PADI4_3, an inhibitor specific for the active conformation of PADI4, PADI4_7, an inert binder, and PADI4_11, an activator. These peptides represent a valuable set of probes to study the structure and function of PADI4, which are already utilised in this study, for example to investigate the role of human PADI4 in NETosis. Regrettably, their high specificity for the human enzyme will preclude their use in mouse models.

Overall, this manuscript describes a substantial body of work and takes advantage of the relatively new RaPID approach. I have a number of comments that should be addressed before acceptance, as follows:

The Introduction is well written and makes a strong case for the development of PADI4 ligands, both inhibitors and activators. Nonetheless, I felt that that it was too long, and could be trimmed to half to 2/3 of its current length without loss of clarity.

Based on this reviewer's suggestion we have cut the introduction by about a third from 1213 words to fewer than 900 words.

Figure S7C shows the results of a CAPA assay indicating that PADI4_3 and PADI4_3i enter cells. PADI4_11 and PADI4_11i are also cell permeable, as documented in Figure S9E. The Discussion (p. 17) acknowledges that PADI4_3 and PADI4_11 show surprisingly high cell permeability, but this important point is not further elaborated. Given the challenges associated with getting peptides and macrocycles across cell membranes, the apparent permeability of these peptides should be further characterised. While this may be beyond the scope of the

current manuscript, the desirability of further work should be mentioned. Is it known if the peptides are taken up initially into endosomes and then released into the cytoplasm?

We thank the reviewer for their insightful comment, and we have addressed their question performing cell penetration assays at 4 °C to assess whether the peptides use active or passive transport to enter the cell (please see updated Supplementary Figure 8D and Supplementary Figure 10G, also included below).

The CP_{50} for PADI4_3 is 7 times higher at 4 °C than at 37 °C (Supplementary Figure 8C,D), while the CP_{50} for PADI4_11 is 4 times higher at 4 °C than at 37 °C (Supplementary Figure 10F,G). Based on these findings, we conclude that the peptides enter the cell through an active transport mechanism.

Figure S8D

Supplementary Figure 8. PADI4_3i does not inhibit PADI4 and PADI4_3 and PADI4_3i are cell permeable and non-toxic. **A.** PADI4_3i does not bind to PADI4. Binding kinetics between PADI4 and PADI4_3i measured by SPR. A representative experiment is shown. This experiment was performed three times with similar results. **B.** PADI4_3i does not inhibit PADI4. COLDER assay with PADI4_3i at different concentrations (10 - 0.0003 μ M) and 10 mM $CaCl_2$. Data are normalised to activity of PADI4 in the presence of 0.1% DMSO vehicle. Data show mean \pm SEM of three independent replicates. Each replicate was performed in triplicate. **C.** PADI4_3 and PADI4_3i enter cells. CAPA assay with PADI4_3 and PADI4_3i. Data show mean \pm SEM of three different experiments. Each replicate was performed in

triplicate. Data are normalised to cells with no peptide treated with TMR dye (Promega) (100 %) and cells with no peptide and no dye (0 %). **D. CAPA assay with PADI4_3 at 4 °C demonstrates that PADI4_3 enters the cells via active cellular transport. Data show mean \pm SEM of two independent experiments. Each replicate was performed in triplicate. Data are normalised as in C.** **E.** Assessment of cytotoxicity by live cell imaging with Incucyte® Cytotox Green Dye, as a measure for cell death. hPADI4 expressing mES cells treated with 1 μ M PADI4_3 or DMSO vehicle (0.1%), and imaged for 24h.

Figure S10G

Supplementary Figure 10. PADI4_11 is selective for hPADI4, is not a PADI4 substrate and is cell permeable. A-D. PADI4_11 and PADI4_12 do not activate hPADI1 (A), hPADI2 (B), hPADI3 (C) or mPADI4 (D). COLDER assays were performed in presence or absence of PADI4_11 or PADI4_12 at 30 μ M and different concentrations of CaCl₂. $K_{50Ca^{2+}}$ is the concentration of CaCl₂ that yields half maximal PADI activity. Data represent mean \pm SEM of three independent replicates. Each replicate was performed in triplicate. Data were normalised against the activity of each PADI in the presence of 0.1% DMSO vehicle and 10 mM CaCl₂. **E.** PADI4_11 is not a PADI4 substrate. COLDER assay performed with 0.4 mM PADI4_11 and BAEE was used as positive control. Data represents mean \pm SEM of three independent replicates. Each replicate was done in triplicate. **F.** PADI4_11 and PADI4_11i are cell

permeable. CAPA assay with PADI4_11 and PADI4_11i. Data show mean \pm SEM of three independent experiments performed in triplicate. Data are normalised with cells with no peptide treated with TMR dye (Promega) (100 %) and cells with no peptide and no dye (0 %). **G. CAPA assay with PADI4_11 at 4 °C demonstrates that PADI4_11 enters the cells via active cellular transport.** Data show mean \pm SEM of three independent experiments performed in triplicate. Data was normalised as in **F**.

We further explored which pathways were involved in the transport. For that we used inhibitors of different proteins involved in the active transport in the cell but could not see any difference in presence or absence of inhibitors, suggesting a complex mode of entrance of the cyclic peptides into the cells. We have not included this additional data in the manuscript as we feel it complicates the manuscript without adding any further insight.

The pull-down results in Figure 6 identify a number of binders of bio-PADI4_7. These are discussed briefly in the current manuscript, but clearly represent an interesting avenue for further study.

We thank the reviewer for recognising this point. Indeed, ongoing and future studies will make use of bio-PADI4_7 to identify cellular interactors of PADI4 under different stimuli and cell states, to probe the regulation and function of cellular PADI4. We expect that this work will form the basis of future manuscripts.

Minor comments:

p. 5: 'Sheared salmon sperm DNA was included in the selection buffers to reduce high levels of non-specific nucleic acid binding to PADI4 which were observed during initial optimisations.' Presumably this is based on literature precedent, in which case a reference should be provided.

Sheared salmon sperm DNA is used routinely in different applications (e.g. ChIP-on-Chip or Northern Blotting) as a blocker of unspecific protein-DNA interactions. In the case of mRNA display, we need to make sure that we don't have any unspecific interactions between our protein of interest and the DNA from the library, so salmon sperm DNA is used as the reagent to block these interactions. We have added a reference from a previous use of sheared salmon sperm DNA to block unspecific interactions in a mRNA display (DOI 10.1074/jbc.M901547200).

p. 7: 'Most notably H640, which...' This is not a sentence.

We have amended this sentence.

p. 9: CAPA should be defined at first use.

We thank the reviewer for their comment and have defined CAPA at its first use.

p. 21: 'Peptides were synthesised using NovaPEG Rink Amide resin as C-terminal amides by standard Fmoc-based solid phase synthesis as previously described...' A reference is needed.

We have added a reference with the method used for the synthesis of the peptides.

p. 21: 'Pure peptides were lyophilised and dissolved in DMSO for further use.' Is this because they were poorly soluble in water?

In general, cyclic peptides identified from RaPID selections show relatively poor solubility in water. We usually dissolve them at high concentration in DMSO and dilute them at the working concentrations for the different assays in water or buffer.

p. 23: 'For colour development, 200 µl of COLDER solution, consisting on 1 volume..' Replace on with of

We have replaced on with of.

p. 25: 'for lysated-based' lysate-

We thank the reviewer for spotting the mistake and have corrected it.

p. 25: 'A minimum of nine images were captured..' was

In this case we think that "were" is the correct form of the verb, as it relates to the plural "images", so we have not amended this sentence.

p. 29: bovine serum albumin should be abbreviated

Bovine serum albumin has been abbreviated on page 29.

Table 1: the heading should indicate to which target the affinities refer.

We thank the reviewer for their comment. We have added the name of the target (Bio-hPADI4 wt) in the heading of Table 1.

Figure 5D: the dark shading makes it difficult to identify the residues; I suggest using lighter shading colours.

We thank the reviewer for their suggestion and we have changed the colours of figures 2F and 5D for lighter colours.

Standard abbreviations for time should be used throughout.

Time units have been changed to standard ones.

Reviewer #2 (Remarks to the Author): changes are highlighted in the manuscript in green.

Bertran et al (2024) A cyclic peptide toolkit reveals mechanistic principles of PAD4 regulation
The manuscript by Bertran et al describes the use of the RAPID approach to identify a series of cyclic peptides that bind selectively to human PAD4. These cyclic peptides fall into one of three categories: i) inhibitors of PAD4 activity; ii) compounds that allosterically decrease the concentration of calcium that is required to activate PAD4; and iii) peptides that bind to PAD4 and can be used to isolate this protein from complex mixtures. All three types of compounds are potentially useful. For example, the authors disclose compound 3 which binds to the active site of PAD4 to inhibit its in vitro activity with an IC₅₀ of ~50 nM. Impressively, the authors obtained the cryoEM structure of PAD4 bound to this cyclic peptide and identified key interactions that are important for its potency and selectivity. The finding that compound 11 allosterically decreases the calcium dependence of PAD4 is also impressive because it provides strong data supporting the notion that an interacting protein can modulate the calcium dependence of the protein to decrease the concentration of calcium required to activate PAD4 to physiologically relevant levels. Although 11 does not decrease the concentration to <1 micromolar the reduction is nonetheless significant. The development of the biotinylated version of compound 7 is also important because it will facilitate the isolation of PAD4 from a variety of complex mixtures. Notably, the authors use this compound to identify potential interacting proteins. Future work will be needed to validate these potential interacting proteins. Overall, this manuscript represents a significant advancement and provides tools that will be used by the wider community.

This reviewer recommends acceptance once the authors address the following.

1. The authors note several times that GSK484 binds the inactive form and that it would be better to have an inhibitor that binds the active form. I think this description of GSK484 is unfair

and could be improved as this compound is an allosteric inhibitor that preferentially binds to the calcium free form of the enzyme, stabilizing it, such that calcium cannot bind and activate the enzyme. Since the calcium free and calcium bound forms of PAD4 are in rapid equilibrium, GSK484 will inhibit PAD4 in the presence of calcium. It is just that at very high levels of calcium less inhibition is observed.

We thank the reviewer for this point and have amended the text accordingly.

2. Several GSK484 derivatives have been disclosed by BMS, Gilead, and Jubilant Therapeutics and this work should be discussed. See Curr. Op. Chem. Biol. 75, 102313 for references.

We thank the reviewer for this point. We have discussed the GSK484 derivatives and added the relevant references.

3. PAD1 and PAD2 specific inhibitors have been disclosed [PMC5477668 Angew Chem Int Ed Engl. 58, 12476]. These works should be referenced.

Thank you for providing these references, we have added them to the Introduction.

4. The PAD community, especially the rheumatology community, routinely refers to the proteins as PADs and the gene as PADI. I know they are annotated as PADI in uniprot but when the vast majority of papers refer to the enzymes as PADs, I think the authors should follow convention.

Thank you for raising this important point. We agree that there is diversity in the nomenclature used for this protein family and this is a wider issue in the field, with some papers referring to the enzymes as PADI and others as PADs. Here, we chose PADI4 to follow the uniprot annotation and also due to the fact that an unrelated gene with the name PAD4 exists in plants (PMID: 10557364). We respectfully suggest keeping this nomenclature for the reasons above but have introduced both abbreviations in the abstract.

5. The authors should determine whether compound 11 is a PAD4 substrate.

We have performed an experiment to determine whether PADI4_11 is a PADI4 substrate.

For that, we used the COLDER assay using PADI4_11 as a substrate and compared the absorbance to the one obtained with the model substrate BAEE. The experiment revealed that PADI4_11 is not a PADI4 substrate.

The following graph was added to Supplementary Figure 10.

6. The authors note that compounds 11 and 12 can activate PAD4 activity by 36% and 12% respectively. Is that increase in activity statistically significant?

We have calculated the statistics for this experiment. The increase in activity is significant for both peptides. For PADI4_11 peptide the p value is <0.0001 whilst the p value for peptide PADI4_12 is 0.03. We have added these values in the text.

7. Fig 2a. GSK488 should be GSK484

We apologise for the mistake and have amended the figure accordingly.

8. Fig 2e. define the color scale in the figure legend.

We thank the reviewer for the comment and have added the colour scale in the figure legend.

9. The authors should highlight the H4R enrichment data and address whether compound 3H4R would be a substrate.

We have performed a COLDER assay to determine if PADI4_3_H4R is a substrate of PADI4.

For that, we used the COLDER assay using PADI4_3_H4R as a substrate and compared the absorbance to that obtained with the model substrate BAEE. The experiment revealed that PADI4_3_H4R is not a PADI4 substrate.

A new supplementary figure, Figure S6 was added to demonstrate this finding.

10. The PADI4_3 nomenclature is a little clumsy. It would be simpler if the authors used bold numbers, e.g. 3, 7, 11, etc to refer to the compounds.

We thank the reviewer for their suggestion and to make nomenclature clearer we have added the numbering of the peptides in bold.

11. Panel 4F appears to quantify the data in 4E but the data does not seem to match up well. For example, with 11i there is a significant signal at 30 μ M that is not apparent in 4F.

We thank the reviewer for their observation and after re-calculating the data we realised that we didn't include the data from one of the experiments on the figure of the first submission.

We have now amended the figure 4E accordingly.

12. The authors used 50 nM of PAD4 in their inhibition assays and their most potent compound has an IC₅₀ of ~ 50 nM. Since they are titrating the enzyme, does the IC₅₀ for 3 underestimate the true IC₅₀? This seems likely based on their SPR data. The authors should comment on this issue in the manuscript.

Thank you for highlighting this issue. We agree that given the concentration of enzyme in our assay is similar to this IC50 we are most likely underestimating the true IC50. As suggested we have added a comment to this effect in the main text:

Note that as the measured IC₅₀ value for PADI4_3 is similar to the concentration of recombinant PADI4 enzyme used in the assay (50 nM) it is likely a slight underestimate. (ref Tonge ACS Infect. Dis. 2019, 5, 796–808)

13. Figure S14 panel G. This panel is unlabeled. Is this wtPAD4 or a specific mutant. Figure S14 panel G corresponds to variant R394A. We apologise for the mistake and have amended the figure accordingly.

14. Regarding the pulldown data with bio7, it would be nice to determine whether any of the identified proteins are truly PAD4 interacting proteins by western blotting. We have added pulldown/Western blot data that show the interaction between bio-PADI4_7-isolated PADI4 and two of the proteins identified by Mass Spec, Bcr and Impdh2. The data represented in Figure S16B show one of two independent experiments.

Please note also that we have modified our Mass Spec data analysis to increase stringency, reporting higher confidence interactors. As we report comparisons between conditions, we have also now used imputation of missing values to include within the analysis proteins that reproducibly enriched in one biological condition but not another. As a result, we have modified the volcano plots in Figures 6B-D. Details of the analysis are described in the Methods and the Discussion section has been modified accordingly. The primary Mass Spec data, with and without imputation, are represented in the updated Supplementary File 2.

Reviewer #3 (Remarks to the Author): changes are highlighted in the manuscript in magenta

The manuscript by Bertran is very well prepared and presented and will make an important contribution to the field. The results are well supported by the data and a broad range of methods have been used to identify and validate the new inhibitors. I am not an expert on the system or the activity assays but they appear to be robust. My experience is within structural biology and the data appear robust and the data processing looks appropriate.

I do not have any significant edits to suggest nor any additional experiments. The one

suggestion i had which would aid in robustness is that the authors should show the data for the structures ie the map. The fitted model is shown but there are no figures which show the correlation of map and model. At the resolution there will be some ambiguity in the fit in parts of the model. Therefore it would be good to have some examples of the correlation between map and model. This does not need to be done for all figures as it can reduce the clarity. For example in Supp figure 13 what is the map quality like in panels B to E?

We thank the reviewer for their suggestion, and we have added different figures that show the electronic density of PADI4_3 (Supplementary Figure 3B), the residues of PADI4 involved in the binding with PADI4_3 (Supplementary Figure 3A), PADI4_11 (Supplementary Figure 14A) and the residues involved in the binding between PADI4 and PADI4_11 (Supplementary Figure 14B).

REVIEWER COMMENTS

Reviewer #2 (Remarks to the Author):

The authors have addressed most of my comments however they have not adequately addressed the following:

Comment 4. The PAD community, especially the rheumatology community, routinely refers to the proteins as PADs and the gene as PADI. I know they are annotated as PADI in uniprot but when the vast majority of papers refer to the enzymes as PADs, I think the authors should follow convention. Response: Thank you for raising this important point. We agree that there is diversity in the nomenclature used for this protein family and this is a wider issue in the field, with some papers referring to the enzymes as PADI and others as PADs. Here, we chose PADI4 to follow the uniprot annotation and also due to the fact that an unrelated gene with the name PAD4 exists in plants (PMID: 10557364). We respectfully suggest keeping this nomenclature for the reasons above but have introduced both abbreviations in the abstract.

Comment: It is unfair to say that some papers use the PAD convention and some use PADI when the vast majority of papers in the field use PAD to refer to the proteins including those published by the original leaders in the field such as Michi Yamada. The use of PADI when referring to the protein is really limited to the chromatin community and that is not universally followed.

Comment 5: The authors should determine whether compound 11 is a PAD4 substrate.

Response: We have performed an experiment to determine whether PADI4_11 is a PADI4 substrate. For that, we used the COLDER assay using PADI4_11 as a substrate and compared the absorbance to the one obtained with the model substrate BAEE. The experiment revealed that PADI4_11 is not a PADI4 substrate. The following graph was added to Supplementary Figure 10.

Comment: The data in figure S10 show that there is a statistically significant increase in the absorbance of PADI4_11 compared to the no substrate control. That data would indicate that the compound is a substrate, albeit one that is relatively weaker than BAEE based on the data in the adjacent column. Ideally, the authors would determine a Km value for PADI4_11. The authors should also convert absorbance units to micromoles of product in Figure S10 and Figure S6.

Reviewer #3 (Remarks to the Author):

The manuscript reads very well and the additional corrections add to the robustness and clarity. I look forward to seeing the paper published it will make an important contribution to the field.

REVIEWER COMMENTS

We thank all reviewers for their positive evaluation of our revised manuscript. We provide a point-by-point response to their outstanding comments in green below.

Reviewer #2 (Remarks to the Author):

The authors have addressed most of my comments however they have not adequately addressed the following:

Comment 4. The PAD community, especially the rheumatology community, routinely refers to the proteins as PADs and the gene as PADI. I know they are annotated as PADI in uniprot but when the vast majority of papers refer to the enzymes as PADs, I think the authors should follow convention.

Response: Thank you for raising this important point. We agree that there is diversity in the nomenclature used for this protein family and this is a wider issue in the field, with some papers referring to the enzymes as PADI and others as PADs. Here, we chose PADI4 to follow the uniprot annotation and also due to the fact that an unrelated gene with the name PAD4 exists in plants (PMID: 10557364). We respectfully suggest keeping this nomenclature for the reasons above but have introduced both abbreviations in the abstract.

Comment: It is unfair to say that some papers use the PAD convention and some use PADI when the vast majority of papers in the field use PAD to refer to the proteins including those published by the original leaders in the field such as Michi Yamada. The use of PADI when referring to the protein is really limited to the chromatin community and that is not universally followed.

Response:

When preparing the manuscript, we took care to use formal nomenclature as this is typically a reporting requirement and therefore used the Uniprot annotation for the peptidylarginine deiminase proteins. In this case, following convention would go against formal nomenclature, yet the subjective nature of convention is exactly the reason why formal nomenclature has been introduced and is insisted upon. We respectfully ask that this request is editorially overridden.

Comment 5: The authors should determine whether compound 11 is a PAD4 substrate.

Response: We have performed an experiment to determine whether PADI4_11 is a PADI4 substrate. For that, we used the COLDER assay using PADI4_11 as a substrate and compared the absorbance to the one obtained with the model substrate BAEE. The experiment revealed that PADI4_11 is not a PADI4 substrate. The following graph was added to Supplementary Figure 10.

Comment: The data in figure S10 show that there is a statistically significant increase in

the absorbance of PADI4_11 compared to the no substrate control. That data would indicate that the compound is a substrate, albeit one that is relatively weaker than BAEE based on the data in the adjacent column. Ideally, the authors would determine a K_m value for PADI4_11. The authors should also convert absorbance units to micromoles of product in Figure S10 and Figure S6.

Response:

Thank you for raising this point. We acknowledge that the increase in absorbance of PADI4_11 relative to no substrate control is statistically significant and have modified the text to highlight this (highlighted in yellow on page 11, and pasted at the end of this response). However, this very low level of activity elicited by PADI4_11 is unlikely to be biologically significant (despite showing statistical significance). It is important to note that any peptide containing an arginine is likely to act as a PADI substrate in *in vitro* assays, in the presence of excess enzyme and excess calcium. PADI4 is also known to autocitrullinate itself (¹⁻³), and this itself would contribute to the signal obtained in this assay, given that PADI4_11 activates PADI4 compared to the no substrate control. Additionally, our CryoEM data of PADI4 binding to the PADI4_11 peptide show that the primary binding site of the peptide is not the active site, but an allosteric site. The fact that PADI4_11 stimulates citrullination when used as a substrate could suggest that it can also bind at the active site, but all activity assays involving PADI4_11 are confounded by the allosteric activating effect of PADI4_11. In this scenario, there is one high affinity binding site (the allosteric site) and a second likely very low affinity binding site (the active site). In activity assays, the PADI4_11 (activating) peptide binding in the first location will be providing some activation which will increase the reaction rate relative to measuring with any other peptide substrate. Therefore, any measurements we obtain here would be confounded by the dual binding, activation and potential autocitrullination and would not be a true reflection of the K_m or k_{cat} . Given that it would be very challenging to measure a meaningful K_m and the fact that an *in vitro* K_m would not add to the conclusions of the manuscript we have not performed this suggested experiment.

As suggested, however, absorbance has been converted to micromoles of citrulline produced in Figure S10 and Figure S6.

Revised text: “We also tested whether PADI4_11 was a substrate of PADI4 (Figure S10E). A small, but statistically significant, increase in citrullination activity was observed in the presence of PADI4_11 (400 μ M) relative to the no substrate control. This may suggest that in addition to acting as an activator, PADI4_11 is also a very poor substrate of PADI4, although the small increase in citrullination activity observed may also be due to activation by PADI4_11 allosteric binding enhancing the amount of PADI4 autocitrullination observed.”

Reviewer #3 (Remarks to the Author):

The manuscript reads very well and the additional corrections add to the robustness and

clarity. I look forward to seeing the paper published it will make an important contribution to the field.

Response

We thank the reviewer for evaluating our manuscript and are pleased they feel it will make an important contribution to the field

References:

1. Andrade, F. *et al.* Autocitrullination of human peptidyl arginine deiminase type 4 regulates protein citrullination during cell activation. *Arthritis & Rheumatism* **62**, 1630–1640 (2010).
2. Darrah, E. *et al.* Citrulline Not a Major Determinant in the Recognition of Peptidylarginine Deiminase 2 and 4 by Autoantibodies in Rheumatoid Arthritis. *Arthritis & Rheumatology* **72**, 1476–1482 (2020).
3. Mondal, S. *et al.* Site-specific incorporation of citrulline into proteins in mammalian cells. *Nat Commun* **12**, 45 (2021).